# PROVABLE LOW-FREQUENCY BIAS OF IN-CONTEXT LEARNING OF REPRESENTATIONS

## ABSTRACT

In-context learning (ICL) enables large language models (LLMs) to acquire new behaviors from the input sequence alone without any parameter updates. Recent studies have shown that ICL can surpass the original meaning learned in the pre-training stage through internalizing the structure of the data-generating process (DGP) of the prompt into the hidden representations. However, the mechanisms by which LLMs achieve this ability are left open. In this paper, we present the first rigorous explanation of such phenomena by introducing a unified framework of double convergence, where hidden representations converge both over context and across layers. This double convergence process leads to an implicit bias towards smooth (low-frequency) representations, which we prove analytically and verify empirically. Our theory explains several open empirical observations, including why learned representations exhibit globally structured but locally distorted geometry, and why their total energy decays without vanishing. Moreover, our theory predicts that ICL has an intrinsic robustness towards high-frequency noise, which we empirically confirm. These results provide new insights into the underlying mechanisms of ICL, and a theoretical foundation to study it that hopefully extends to more general data distributions and settings.

## 1 INTRODUCTION

It has become a major challenge for today's machine learning community to understand how large language models (LLMs) perform in-context learning (ICL), i.e. the ability to learn patterns or tasks solely from input sequences without any gradient updates (Brown et al., 2020; Min et al., 2022a; Garg et al., 2022; Akyürek et al., 2022a). Empirical studies have demonstrated that LLMs can carry out a variety of tasks, including logical reasoning (Wei et al., 2022), programming (Gao et al., 2023), and solving mathematical problems (Hendrycks et al., 2021); recent work also suggests that LLMs are able to stay robust against noisy prompts (Cheng et al., 2025; Alazraki et al., 2025). However, the mechanisms underlying these capabilities remain largely elusive. A particularly striking phenomenon, recently highlighted by Park et al. (2024), shows that when a pre-trained LLM is fed a sequence generated by a random walk on a planar graph, with each node corresponds to a word (see "Data Generating Process" part of Figure 1), the model's hidden representations converge to a state that reflects the original graph structure, even though the graph itself was never explicitly provided. We refer to this emergent behavior as In-Context Learning of Representations (ICLR).

The ICLR phenomenon suggests an important mechanism of ICL: the model (in-contextly) learns to embed the information of the data-generating process (DGP) into the hidden representations. Therefore, understanding the ICLR phenomenon is a crucial step towards a deeper understanding of the mechanisms of ICL. Moreover, Park et al. (2024) also shows that an energy function decays over the course of this process, suggesting there might be an underlying principle that drives the emergence of ICLR. However, it is left open what the nature of this principle is and how it applies to the representations.

In this paper, we present the first theoretical explanation of the ICLR phenomenon, showing that it arises as a consequence of an intrinsic bias in LLMs towards low-frequency hidden representations. The central idea of our theoretical framework is a process we call **Double Convergence**, wherein hidden representations converge both along the context length and across layers. Specifically, the low-frequency bias emerges from the interaction of the **Context-wise Process** and the **Layer-wise Process**.

1. **Context-wise Process**: If the attention map "reflects" a function that depends only on token identities (formally defined in Definition 2 and Definition 3), and the representations converge to a set of *limiting representations* as the context length increases (formally defined in Definition 1), then the attention effectively applies the reflected function to the limiting representations. This result is formally shown in Theorem 1;

2. **Layer-wise Process**: The *limiting representations* across layers, and eventually converge to a state that captures the distributional properties of the input sequence. Under the specific DGP considered in Park et al. (2024), we prove that this convergence exactly yields the representations characteristic of the ICLR phenomenon. This result is formally stated in Theorem 2.

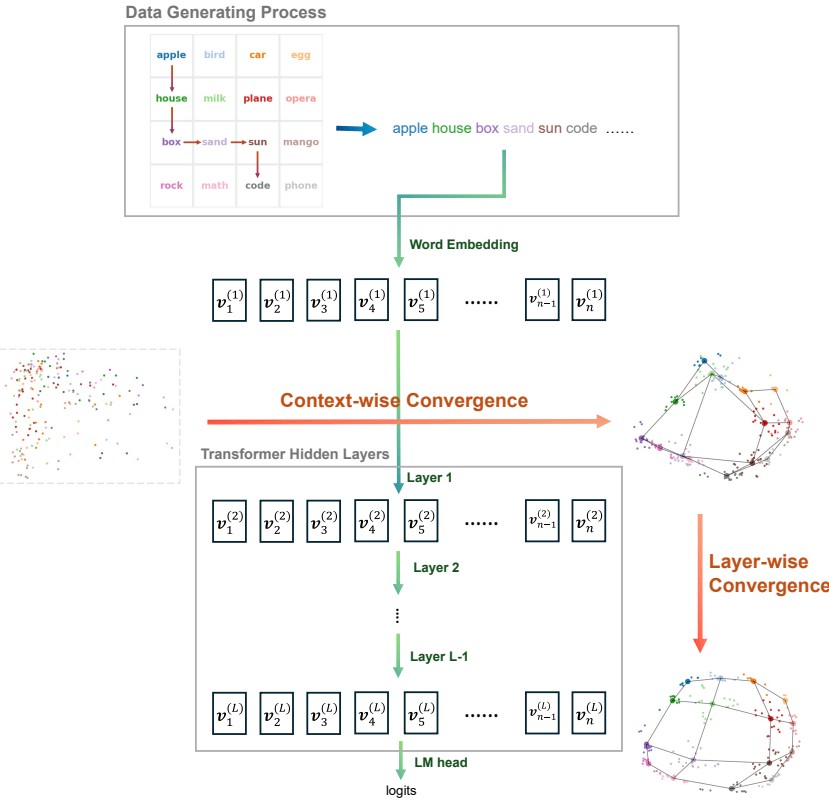

Figure 1: **Overview of the DGP and the Double Convergence Process.** The input sequence is generated by a random walk over a graph defined on the vocabulary (top), and then passed into a pre-trained Transformer model. **Context-wise convergence** occurs within each layer: token representations associated with the same word converge into tight clusters. **Layer-wise convergence** occurs across layers: the cluster centers gradually evolve towards a structure aligned with the underlying graph.

The concept of Double Convergence is illustrated in Figure 1. Our theoretical framework provides a comprehensive explanation for several phenomena and open questions raised in Park et al. (2024). Furthermore, as both a validation and application of our theoretical results, we also demonstrate that ICL exhibits implicit robustness against high-frequency noise in the input data, which is consistent with recent empirical findings (Cheng et al., 2025; Alazraki et al., 2025). While our main analysis focuses on the specific DGP used in Park et al. (2024), the techniques and theoretical insights we develop are able to be extended beyond this particular setting. In Appendix E, we provide a generalized framework that is decoupled from any specific DGP, highlighting the broader applicability of our results.

In summary, our main contributions in this paper are as follows:

1. We identify the double convergence process, which serves as a general framework for studying the evolution of the representation in ICL (Sections 3 and 4);

2. We provide theoretical explanations for several previously unexplained phenomena in Park et al. (2024), including why ICL can suppress the original meaning of each word learned in pre-training (Section 5.1), why the learned representations form an apparently regular yet slightly distorted structure (Sections 5.2 and 5.3), and why the energy decreases but does not converge to zero (Section 5.4);

3. Our theory highlights an implicit low-frequency bias in ICL, and predicts that LLMs are naturally robust to errors in the input prompts, which we have verified empirically (Section 5.5).

This paper is organized as follows. In Section 2, we define notations used and the problems considered throughout the paper, as well as outline our method. In Section 3, we present our results on context-wise convergence, and in Section 4, we establish the main result by combining context-wise and layer-wise convergence. In Section 5, we discuss the theoretical implications of our framework and address specific empirical observations. Finally, we conclude in Section 6 with a summary and outlook for future work. Due to space constraints, we defer some of the content, such as a review of related work, the formal proofs of the theoretical results and the empirical validation of the assumptions, to the appendix.

## 2 PRELIMINARIES

Throughout this paper, we use bold upper-case letters to represent matrices (e.g. $\boldsymbol{A}$), bold lower-case letters to represent vectors (e.g. $\boldsymbol{x}$) and calligraphic upper-case letters to represent sequences of vectors (e.g. $\mathcal{V} = \{\boldsymbol{v}_k\}_{k=1}^n$). For a matrix or a vector, we use plain lower-case letters to represent their entries (e.g. $a_{i,j}$ represents the $i,j$-th entry of $\boldsymbol{A}$). For a number $n \in \mathbb{N}$, we denote $\{1, 2, \cdots, n\}$ by $[n]$. For a set $S$, we use $2^S$ to represent its power set. We use $\mathbf{1}$ to represent a vector with all entries being 1, whose dimensionality is inferred from context. For a logical statement $\phi$, we define $\mathbb{1}_{\{\phi\}} = \begin{cases} 1 & \phi \text{ is true} \\ 0 & \phi \text{ is false} \end{cases}$ to be its indicator function. Given a sequence of vectors $\mathcal{Z} = \{\boldsymbol{z}_x\}_{x=1}^c \in (\mathbb{R}^d)^c$, we define $\boldsymbol{Z} = \text{mat}\, \mathcal{Z} \in \mathbb{R}^{d \times c}$ to be the matrix formed by column-wise stacking of the vectors in $\mathcal{Z}$, i.e. the $i$-th column of $\boldsymbol{Z}$ is $\boldsymbol{z}_i$. Given a matrix $\boldsymbol{A} \in \mathbb{R}^{n \times n}$ and a sequence of vectors $\mathcal{V} = \{\boldsymbol{v}_k\}_{k=1}^n \in (\mathbb{R}^d)^n$, let $\boldsymbol{A}\mathcal{V}$ to be a sequence defined as $\boldsymbol{A}\mathcal{V} = \left\{ \sum_{j=1}^n a_{k,j} \boldsymbol{v}_j \right\}_{k=1}^n$. For a vector function $\sigma : \mathbb{R}^d \to \mathbb{R}^d$ and a matrix $\boldsymbol{Z} \in \mathbb{R}^{c \times d}$, we use $\sigma(\boldsymbol{Z})$ to represent applying $\sigma$ column-wisely to $\boldsymbol{Z}$.

### 2.1 DATA GENERATION

Throughout this paper, we assume the input is a sequence of tokens $\mathcal{X} = \{x_k\}_{k=1}^n \in [c]^n$, where $n$ is the sequence length and $c$ is the vocabulary size (each token is simply a number in $[c]$). We assume that the sequence is sufficiently long, i.e., $n > 10c$.

Let $\mathcal{G} = ([c], \mathcal{E})$ be a connected undirected graph defined over the vocabulary (with each node being a word). Let $\widetilde{\boldsymbol{W}} = \{\widetilde{w}_{x,y}\}_{x,y \in [c]}$ denote the adjacency matrix of $\mathcal{G}$, and let $\boldsymbol{\pi} = \{\pi_x\}_{x \in [c]} \in \mathbb{R}^c$ be the stationary distribution $\mathcal{G}$ (i.e. $\boldsymbol{\pi}$ is the $L_1$ normalized Perron vector of $\widetilde{\boldsymbol{W}}$).

We define the *reweighted adjacency* matrix $\boldsymbol{W} = \{w_{x,y}\}_{x,y \in [c]}$ as $w_{x,y} = \widetilde{w}_{x,y} \pi_x \pi_y$. Note that $\boldsymbol{W}$ is also a non-negative symmetric matrix, and can therefore be viewed as the adjacency matrix of a reweighted version of $\mathcal{G}$. Define the (reweighted) degree vector $\boldsymbol{d} = \{d_x\}_{x \in [c]} \in \mathbb{R}^c$ as $d_x = \sum_{y \in [c]} w_{x,y}$. Let $\boldsymbol{D} = \text{diag}(\boldsymbol{d})$ be the degree matrix of $\boldsymbol{W}$.

We assume the input sequence $\mathcal{X}$ satisfies the following data-generating process: the first $c$ tokens are fixed as a traversal of the vocabulary (i.e. $x_k = k$ for $k \in [c]$), and starting from $x_{c+1}$, the remaining tokens are generated by a random walk on the graph $\mathcal{G}$, with the initial token $x_{c+1}$ being sampled from the stationary distribution $\boldsymbol{\pi}$ (i.e. the probability of $x_{c+1} = y$ is $\pi_y$)[1]. For any word $x \in [c]$ and position $k \in [n]$, let $F_{x,k}$ be the frequency of token $x$ in the first $k$ elements of the sequence, i.e. $F_{x,k} = \sum_{j=1}^k \mathbb{1}_{\{x_j = x\}}$.

---

[1]This DGP is essentially the same as Park et al. (2024). We fix the first $c$ tokens only to avoid trivial but complicated edge cases in the analysis, and since we study the asymptotic behavior of the model, the effect of the first a few tokens is negligible.

## 2.2 MODEL ARCHITECTURE

Throughout the paper, we consider a simplified yet deep and non-linear Transformer model, as described in Algorithm 1, where $d \in \mathbb{N}$ is the hidden and input dimension, $L \in \mathbb{N}$ is the number of layers, $\boldsymbol{A}^{(\ell)} = \left\{ a_{k,j}^{(\ell)} \right\}_{k,j \in [n]}$ is the (single-head) attention map at layer $\ell$, and $\sigma^{(\ell)} : \mathbb{R}^d \to \mathbb{R}^d$ is the neuron-wise transformations at layer $\ell$ (which may include feedforward networks (FFNs), normalization layers, and other non-linearities).

The most critical simplification in our model is that we treat the attention maps as given, instead of being generated from hidden representations. Our assumed attention structure, as discussed in Appendix A, can be viewed as a generalization of induction heads, whose existence has been broadly verified in practice (Olsson et al., 2022). In exchange for this simplification, we are able to explicitly characterize the structure of the hidden representations in a deep and non-linear model, enabling us to rigorously explain multiple in-context learning behaviors. This contrasts with prior works that remain entangled in the complexity of layer-wise interactions in full Transformer architectures (Yang et al., 2022). Moreover, we validate our assumption empirically on real models in Appendix G, finding that our assumptions on self-attention effectively explain more than 70% of actual attention connections, indicating its practical justifiability.

This model described in Algorithm 1 also omits several other standard components such as normalizations and residual connections. This is to not over-complicate the theoretical results while still capturing the core mechanisms and challenges of the model. There is flexibility in our analysis to include other components, but we choose to focus on a minimal version in the main paper to clearly highlight the key ideas behind our theoretical results. See Appendix F for further discussion on integrating additional components.

## 2.3 METHODOLOGY OUTLINE

As noted before, we treat the attention maps in the Transformer as given, rather than dynamically generated from the hidden representations. Specifically, we assume the attention maps in the model are composed of a class of structured maps we refer to as *balanced attentions*, formally defined in Definition 3. These are attention maps whose connectivity patterns are determined by a function of the input tokens. This concept can be viewed as a generalization of the notion of induction heads (see eq. (11)), as illustrated by the structural similarity between eq. (11) and eq. (3).

---

**Algorithm 1** Transformer forward process

**input**: $\left\{ \boldsymbol{v}_k^{(1)} \right\}_{k=1}^n \in \left( \mathbb{R}^d \right)^n$ as $\boldsymbol{v}_k^{(1)} = \boldsymbol{b}_{x_k}$.
**for** $\ell = 1, 2 \cdots L - 1$ **do**
    **for** $k = 1, 2 \cdots n$ **do**
        $\boldsymbol{u}_k^{(\ell)} = \sum_{j=1}^k a_{k,j}^{(\ell)} \boldsymbol{v}_j^{(\ell)};$
        $\boldsymbol{v}_k^{(\ell+1)} = \sigma^{(\ell)} \left( \boldsymbol{u}_k^{(\ell)} \right).$
    **end for**
**end for**
**output**: $\left\{ \boldsymbol{v}_k^{(L)} \right\}_{k=1}^n.$

---

Given this assumption, we are able to prove that the there is a *double convergence* process in the inference dynamics: 1) within each layer, the hidden representations converge to a set of "limiting representations" that only encodes token identity and does not have position information; 2) then, each layer of balanced attentions operates as a transformation on these limiting representations. Consequently, the limiting representations evolve and converge progressively across layers.

Furthermore, as the hidden representations can be viewed as having two axes, which we call the neuron axis and token axis, and the double convergence happens in the token axis, it can be shown that any transformations in the neuron axis, as long as they are well-conditioned, will not affect this double convergence process. This observation allows us to "insert" any neuron-wise transformations (such as FFNs and normalizations) between the self-attention layers without affecting the double convergence behavior, enabling a modular and robust theoretical framework.

## 3 CONTEXT-WISE CONVERGENCE

We begin by establishing a general result: if the attention map reflects a specific function of the underlying tokens, and the representations converge to a set of limiting representations, then the output of the attention layer also converges, towards a transformed set of limiting representations. We start the presentation of this result by defining limiting representations and balanced attention maps.

**Definition 1.** *For a sequence of d-dimensional vectors $\mathcal{U} = \{\boldsymbol{u}_k\}_{k=1}^n \in \left(\mathbb{R}^d\right)^n$, if there exists a number $\gamma > 0$ and another sequence $\mathcal{Z} = \{\boldsymbol{z}_x\}_{x \in [c]}$, such that*

$$\forall k \in [n], \|\boldsymbol{u}_k - \boldsymbol{z}_{x_k}\| \leq \frac{\gamma}{\sqrt{k}}, \tag{1}$$

*then we say $\mathcal{U}$ is a **stable sequence** converging to $\mathcal{Z}$ with parameter $\gamma$, and $\mathcal{Z}$ is the **limiting representation** of $\mathcal{U}$.*

The concept of limiting representations captures the idea that the hidden representation for each token converges to a vector determined solely by the token identity and independent of position.

**Definition 2.** *If a matrix $\boldsymbol{A} = \{a_{k,j}\}_{k,j \in [n]} \in \mathbb{R}_+^{n \times n}$ is lower-triangular and row-stochastic, i.e. it satisfies $a_{k,j} = 0$ for all $j > k$, and $\sum_{j=1}^k a_{k,j} = 1$ for all $k \in [n]$, then we say $\boldsymbol{A}$ is an **attention map**. Moreover, for an attention map $\boldsymbol{A}$, if there exists a scalar $\psi > 0$, such that*

$$\forall k \in [n], j \in [k], \sum_{i=1}^j a_{k,i} \leq \frac{\psi j}{k}, \tag{2}$$

*then we say $\boldsymbol{A}$ is a **balanced attention map** with parameter $\psi$.*

Balanced attention maps are attention maps with a soft uniformity and locality: they prevent the attention from overly concentrating disproportionately on early tokens.

**Definition 3.** *If $\boldsymbol{A} \in \mathbb{R}^{n \times n}$ is a balanced attention map with parameter $\psi$, and there is a function $f : [c] \to 2^{[c]}$, such that for all $k > c$ and $j \in [k]$,*

$$a_{k,j} > 0 \implies x_j \in f(x_k), \tag{3}$$

*and moreover,*

$$\forall k \in [n], \forall y \in f(x), \left| \sum_{j \in [k]} a_{k,j} \mathbb{1}_{\{x_j = y\}} - \frac{F_{y,k}}{\sum_{y' \in f(x)} F_{y',k}} \right| \leq \frac{\psi}{\sqrt{k}} \tag{4}$$

*then we say $\boldsymbol{A}$ **reflects** the function $f$.*

Intuitively, a balanced attention that reflects a function matches a functionally defined neighborhood of the current token, which can be viewed as a generalized notion of induction heads. For example, if $f$ selects all the neighbors of a token, then it can be viewed as an induction head (See Appendix A for more discussion about it). Moreover, a balanced attention that reflects a function also requires attention weights to be distributed roughly proportionally among all words.

Notice that in Definition 3, the $k$-th row of the attention map $\{a_{k,j}\}_{j=1}^n$ is only well defined only there exists $j \in [k]$ such that $x_j \in f(x_k)$ (otherwise all attention weights in this row are 0, violating the assumption that each row of $\boldsymbol{A}$ sums up to 1). This is guaranteed under our setup because we have explicitly set the first $c$ tokens to be a traversal over the entire vocabulary.

We are now ready to state the main theorem of this section. Notice that this theorem stated here depends on the DGP, as it relies on the distribution of $F_{x,k}$. However, it is possible to prove a weaker but more general version of this theorem that is independent of the DGP. See Appendix E for more details.

**Theorem 1.** *There exists a constant $C > 0$ that only depends on $\mathcal{G}$, and an event with probability at least $0.999$, such that within this event, the following statement holds. Suppose $\mathcal{V} = \{\boldsymbol{v}_k\}_{k=1}^n \in \left(\mathbb{R}^d\right)^n$ is a stable sequence converging to $\mathcal{Z} = \{\boldsymbol{z}_x\}_{x \in [c]}$ with parameter $\gamma$, and $\boldsymbol{A} = \{a_{k,j}\}_{k,j \in [n]} \in \mathbb{R}^{n \times n}$ is a balanced attention map with parameter $\psi$ that reflects a function $f : [c] \to 2^{[c]}$. Then $\boldsymbol{A}\mathcal{V}$ is a stable sequence converging to*

$$\mathcal{Z}' = \left\{ \frac{\sum_{y \in f(x)} \pi_y \boldsymbol{z}_y}{\sum_{y \in f(x)} \pi_y} \right\}_{x \in [c]} \tag{5}$$

*with parameter $C\psi(\gamma + N) + C \log n$, where $N = \max_{y \in [c]} \|\boldsymbol{z}_y\|$.*

Theorem 1 shows that applying a balanced attention map that reflects a function to a stable sequence yields another stable sequence, and that the attention operation effectively acts on the limiting representations. In other words, attention maps of this kind preserve convergence and transform limiting representations in a token-consistent way.

# 4 LAYER-WISE CONVERGENCE

In Theorem 1, we showed how limiting representations evolve under a single attention layer. In this section, we study how these limiting representations change across layers.

**Attention Maps.** To analyze layer-wise convergence, we must have a more specific assumption on what exactly the functions are that the attention maps reflect. Specifically, we assume each attention map is a weighted combination of four basic types of maps: $\boldsymbol{A}^{(\ell,A)}$, $\boldsymbol{A}^{(\ell,B)}$, $\boldsymbol{A}^{(\ell,O)}$ and $\boldsymbol{A}^{(T)}$, that satisfies the following assumptions respectively:

1. A-type (self connections): $\boldsymbol{A}^{(\ell,A)}$ is a balanced attention map with parameter $\psi_A^{(\ell)}$ that reflects $f_A : x \mapsto \{x\}$;

2. B-type (neighbor connections): $\boldsymbol{A}^{(\ell,B)}$ is a balanced attention map with parameter $\psi_B^{(\ell)}$ that reflects $f_B : x \mapsto \{y \in [c] | \widetilde{w}_{x,y} > 0\}$;

3. O-type (other connections): $\boldsymbol{A}^{(\ell,O)}$ is a balanced attention map with parameter $\psi_O^{(\ell)}$ that reflects $f_O : x \mapsto [c]$;

4. T-type (trivial connections, i.e. attention sinks): $\boldsymbol{A}^{(T)}$ satisfies $a_{i,j}^{(T)} = 1$ only when $j = 1$.

We assume the attention map at layer $\ell$, i.e. $\boldsymbol{A}^{(\ell)}$, takes the form

$$\boldsymbol{A}^{(\ell)} = \rho_A^{(\ell)} \boldsymbol{A}^{(\ell,A)} + \rho_B^{(\ell)} \boldsymbol{A}^{(\ell,B)} + \rho_O^{(\ell)} \boldsymbol{A}^{(\ell,O)} + \rho_T^{(\ell)} \boldsymbol{A}^{(T)}, \tag{6}$$

where $\rho_\tau^{(\ell)} \geq 0$ ($\tau \in \{A, B, O, T\}$) is the weight of the $\tau$-th type connections, and $\sum_{\tau \in \{A,B,O,T\}} \rho_\tau^{(\ell)} = 1$. Empirically, we find that these four types explain over 70% of attention connections in real models (see Appendix G for details), highlighting the empirical soundness of this classification.

**The Role of FFN.** To enable meaningful layer-wise convergence results with neuron-wise transformations involved, we also need assumptions on the nonlinearity $\sigma^{(\ell)}$ applied at each layer. We introduce the following concept.

**Definition 4.** *For a set $\mathscr{Z} \subseteq \mathbb{R}^{d \times c}$, an orthonormal matrix $\boldsymbol{U} \in \mathbb{R}^{cc \times p}$ and scalars $\gamma_1, \gamma_2 > 0$, if a function $\sigma : \mathbb{R}^d \to \mathbb{R}^d$ satisfies that for any matrix $\boldsymbol{Z} \in \mathscr{Z}$,*

$$\gamma_1 \left\| \boldsymbol{Z} \boldsymbol{D}^{1/2} \boldsymbol{U} \right\| \leq \left\| \sigma(\boldsymbol{Z}) \boldsymbol{D}^{1/2} \boldsymbol{U} \right\| \leq \gamma_2 \left\| \boldsymbol{Z} \boldsymbol{D}^{1/2} \boldsymbol{U} \right\|, \tag{7}$$

*then we say $\sigma$ is a **great mapping** w.r.t. $(\gamma_1, \gamma_2, \mathscr{Z}, \boldsymbol{U})$.*

Intuitively, a great mapping is well-behaved (e.g. smooth) along a given subspace at the point of the limiting representations. This allows us to insert FFNs or other neuron-wise transformations without disrupting convergence.

## 4.1 MAIN RESULTS

Before stating the main results, we first define the concept of the eigenvalue separation ratio of a matrix.

**Definition 5.** *For a symmetric matrix $\boldsymbol{M} \in \mathbb{R}^{c \times c}$ and index $q \in [c]$, let its eigenvalues be $\{\lambda_k\}_{k=1}^c$, arranging in a non-increasing order of absolute values, define $\delta_q(\boldsymbol{M}) = \left| \frac{\lambda_q}{\lambda_{q+1}} \right|$ be the **eigenvalue separation ratio** of $\boldsymbol{M}$.*

With the above assumptions and definitions, we are ready to present our main theorem.

**Theorem 2.** *There exists an event with probability at least $0.99$ such that the following statement holds. Let $n_{[x]}$ be the largest $k \in [n]$ such that $x_k = x$. Let $\boldsymbol{M} = \boldsymbol{D}^{-1/2} \boldsymbol{W} \boldsymbol{D}^{-1/2}$. Let the eigen-decomposition of $\boldsymbol{M}$ be*

$$\boldsymbol{M} = \begin{bmatrix} \boldsymbol{f}_1 & \boldsymbol{X} & \boldsymbol{Y} \end{bmatrix} \begin{bmatrix} \lambda_1 & & \\ & \boldsymbol{\Lambda} & \\ & & \boldsymbol{\Lambda}' \end{bmatrix} \begin{bmatrix} \boldsymbol{f}_1^\top \\ \boldsymbol{X}^\top \\ \boldsymbol{Y}^\top, \end{bmatrix} \tag{8}$$

*where the eigenvalues are arranged in a non-increasing order of absolute values; $\boldsymbol{\Lambda}$ contains $q - 1$ eigenvalues and $\boldsymbol{\Lambda}'$ contains $c - q$ eigenvalues. Let $\boldsymbol{A}^{(\ell)}$ be defined as in eq. (6), and $\left\{ \boldsymbol{v}_k^{(\ell)} \right\}_{k=1}^n$*

*be defined as in Algorithm 1. Suppose each $\sigma^{(\ell)}$ is a great mapping w.r.t. $\left(\gamma_1^{(\ell)}, \gamma_2^{(\ell)}, \mathbb{R}^d, \boldsymbol{X}\right)$ and $\left(\gamma_1'^{(\ell)}, \gamma_2'^{(\ell)}, \mathbb{R}^d, \boldsymbol{Y}\right)$, and have finite Lipschitz constant. Suppose there exists $\epsilon > 0$ satisfying that $\delta_q \left(\rho_A^{(\ell)} \boldsymbol{I} + \rho_B^{(\ell)} \boldsymbol{M}\right)^{-1} \frac{\gamma_2'^{(\ell)}}{\gamma_1^{(\ell)}} \le 1 - \epsilon$ for all $\ell \in [L]$. Then*

$$\lim_{L \to \infty} \lim_{n \to \infty} \frac{\left\| \boldsymbol{V}_n^{(L)} \boldsymbol{D}^{1/2} \boldsymbol{Y} \right\|}{\left\| \boldsymbol{V}_n^{(L)} \boldsymbol{D}^{1/2} \boldsymbol{X} \right\|} = 0, \tag{9}$$

*where $\boldsymbol{V}_n^{(L)} = \mathrm{mat} \left\{ \boldsymbol{v}_{n_{[x]}}^{(L)} \right\}_{x \in [c]}$.*

## 5 DISCUSSION ABOUT THE THEORETICAL RESULTS

In Theorem 2, we proved that the representations converge to the top eigenspace of $\boldsymbol{M} = \boldsymbol{D}^{-1/2} \boldsymbol{W} \boldsymbol{D}^{-1/2}$. This matrix has been extensively studied in the literature on graph learning and spectral methods (Kipf & Welling, 2016; Li et al., 2018; Wu et al., 2019; Yang et al., 2021; Spielman, 2019; Trevisan, 2013). Specifically, since $\boldsymbol{W}$ is a symmetric matrix with non-negative entries, it defines a (weighted) undirected graph. Let $\widehat{\boldsymbol{L}}$ be the symmetrically normalized Laplacian matrix of this graph (Spielman, 2019), it holds that $\boldsymbol{M} = \boldsymbol{I} - \widehat{\boldsymbol{L}}$, i.e. $\boldsymbol{M}$ and $\widehat{\boldsymbol{L}}$ share the same set of eigenvectors, with the order of eigenvalues being reversed. Thus, **top eigenvectors** of $\boldsymbol{M}$ correspond to **low eigenvalues** of $\widehat{\boldsymbol{L}}$, which is known to encode the low-frequency (smooth) and low-energy signals on the graph, as they tend to assign similar values to adjacent nodes. These low eigenvectors of $\widehat{\boldsymbol{L}}$ are often used as coordinates for graph visualization, as it is known that they form figures that match human intuition (Tutte, 1963)[2], and exactly explains why the ICLR phenomenon, where the hidden representations encode global graph structure, emerges in such settings.

We confirm this theoretical prediction by reproducing the experiments in Park et al. (2024) and compare the principal components of the actual hidden representations and the analytical prediction, i.e. top eigenvectors of $\boldsymbol{W}$. The result is shown in Figure 2. It is clear that the analytical predictions align closely with the empirical principal components. All experiments in this paper are done with the pretrained Llama-3.1-8B model, accessed through the NNSight Library (Fiotto-Kaufman et al., 2024).

### 5.1 HOW DOES ICL SUPPRESS ORIGINAL SEMANTIC MEANING?

One of the most surprising observations in Park et al. (2024) is that, under their proposed DGP, ICL can produce word embeddings that no longer reflect the original semantic meaning of each word, but instead align solely with the structure imposed by the DGP. Our theory provides a natural explanation for this phenomenon: the "meaning" encoded in a word representation can be seen as a combination of multiple frequency components. However, as both the model depth and sequence length increase, higher-frequency components are progressively suppressed through the double convergence process. As a result, the semantic features associated with these higher-frequency components are effectively erased, and the representation becomes increasingly dominated by the low-frequency structure induced by the DGP.

### 5.2 WHY START FROM THE 2ND EIGENVECTOR?

In both Theorem 2 and Figure 2, we intentionally omit the first eigenvector of $\boldsymbol{M}$. This is due to the coincidental alignment between the Laplacian and PCA. In short, the 1st eigenvector of $\boldsymbol{M}$ corresponds to a constant vector added to each $\boldsymbol{z}_x'$; however, PCA involves a **centralization step** that removes the mean component from the data. Specifically, it can proved that the first eigenvector of $\boldsymbol{M}$ is exactly $\boldsymbol{d}^{-1/2}$ (Spielman, 2019), and the centralization operation is to projecting $\boldsymbol{V}_n^{(L)}$ onto the space orthogonal to $\boldsymbol{1}$, which is equivalent to projecting $\boldsymbol{V}_n^{(L)} \boldsymbol{D}^{1/2}$ onto the space orthogonal to $\boldsymbol{d}^{-1/2}$, effectively removing the component aligned with the first eigenvector of $\boldsymbol{M}$.

### 5.3 WHY ARE PERIPHERAL NODES COMPRESSED?

In Park et al. (2024), the authors keenly observed that the empirical PCA results (say, for the grid graph as in Figure 2(c)), despite roughly showing the underlying grid structure, appears slightly

---

[2]Likely because humans also have a low-frequency bias in visual processing.

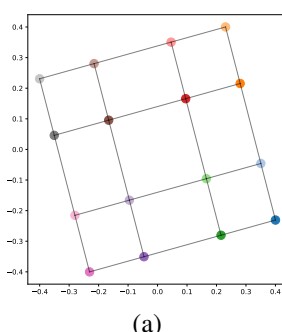 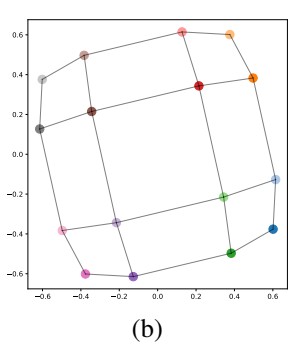 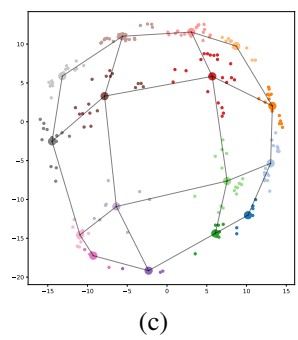

(a)        (b)        (c)

Figure 2: **Comparison between empirical observations and theoretical predictions. (a)**: The 2nd and 3rd eigenvectors of $\widetilde{W}$, illustrating the idealized grid structure hypothesized in Park et al. (2024); **(b)**: The 2nd and 3rd eigenvectors of $M$, as predicted by Theorem 2; **(c)**: The first two principal components of the actual hidden representations from a pre-trained Transformer (Llama-3.1-8B), collected at context positions 2360-2560. In all subfigures, each point represents the limiting representation of a word in the vocabulary. The $x$- and $y$-axes represent the values of the 2nd and 3rd eigenvectors (or the 1st and 2nd principal components), respectively. A small rotation was applied to panels (a) and (b) to better visually align them with (c). This is valid since the 2nd and 3rd eigenvalues of $W$ are equal, and their eigenspace is invariant under rotation.

compressed near the periphery, compared to the actual grid formed by the eigenvectors of the original graph (as in Figure 2(a), or Figure 7 in Park et al. (2024)). The authors in Park et al. (2024) attributed this distortion to uneven visitation frequencies in the random walk:

> *... due to lack of periodic boundary conditions, concepts that are present in the inner 2×2 region of the grid are visited more frequently during a random walk on the graph, while the periphery of the graph has a lower visitation frequency.*

While there is indeed a difference in visitation frequency, we argue that it is not the most fundamental explanation. The true cause lies in the context-wise process. As shown in Theorem 1, the transformation applied by the attention map to the limiting representations is modulated by the stationary distribution $\pi$. As a result, the actual graph the model is aware of is the reweighted graph $W$ instead of the original one $\widetilde{W}$. The top eigenvectors are twisted a little bit according to node degrees since it is reweighted by $\pi$.

### 5.4 WHY ENERGY DECREASES BUT DOESN'T VANISH?

In Park et al. (2024), the authors hypothesized that the structure of the representation is a consequence of energy minimization. While this observation aligns with empirical trends, we argue that energy decay is not the fundamental cause, since 1) it doesn't explain why the model follows the principle of energy decaying, and 2) the energy does not actually converge to 0, despite the 0-energy solutions are actually easy to find. Instead, both the energy decay and the structured representations are consequences of the double convergence.

Formally, let $\Pi_i$ be the projection operator onto the $i$-th eigenspace of $M$, the energy of the limiting representation $\{z_x\}_{x \in [c]}$ under the graph $W$ can be decomposed as follows:

$$\sum_{x,y \in [c]} w_{x,y} \|z_x - z_y\|^2 = \sum_{i=1}^{c} \sum_{x,y \in [c]} w_{x,y} \|\Pi_i(z_x - z_y)\|^2 . \tag{10}$$

As shown in Theorem 2, the projections of the limiting representations onto the low eigenspaces converge to $\mathbf{0}^3$, and thus $\|\Pi_i(z_x - z_y)\|$ converges to $0$ for large $i$. The energy decay is thus a consequence of the representation leaving the corresponding eigenspace.

On the other hand, for the top eigenspaces of $M$ (i.e. small $i$), the projection components persist or even grow. This explains why the total energy does not decay to zero: the representation is leaving high-frequency eigenspaces, but accumulating energy in low-frequency ones.

---

[3]In principle, Theorem 2 is a relative result. However, with a similar proof one can show that the numerator also actually converge to 0 as long as the corresponding eigenvalues are significantly smaller than 1.

We validate this explanation empirically in Figure 3 by calculating the normalized energy along the direction of each component (only the first 5 components are presented for clarity). The energy is computed by taking hidden representations over context positions 2360–2560 and then computing the average representation for each token. From the results, it is clear that while the overall energy decreases across layers, the energy in low-frequency directions (e.g., Component 1 and 2) increases, confirming our theoretical prediction: energy decay arises from the attenuation of high-frequency components, whereas the persistence of low-frequency components prevents the total energy from converging to 0.

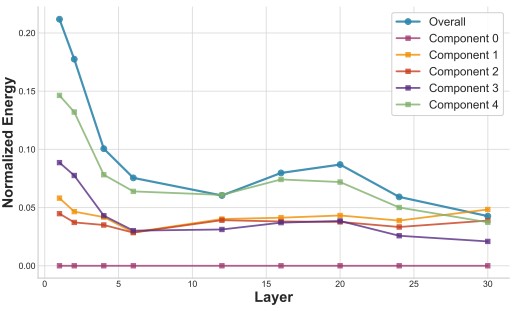 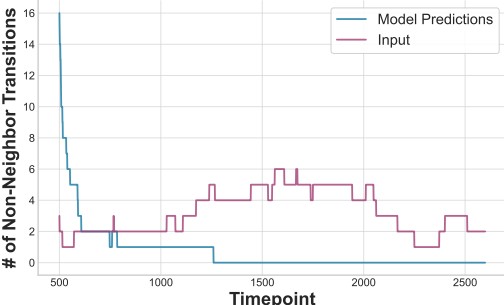

Figure 3: **The Normalized energy evolution across layers.** Each "Component" curve shows the energy along a direction defined by the $k$-th eigenvector of $\boldsymbol{M}$. To eliminate scale effects, the matrix $\boldsymbol{Z}^{(\ell)}$ is normalized to have unit Frobenius norm at each layer.

Figure 4: **Predicted Robustness Against Input Noise.** We inject 1% random noise into the input sequence and plot the number of non-neighbor transitions within a sliding window of the last 500 tokens.

## 5.5 Predicted Robustness Against Noise

Our theoretical framework predicts that high-frequency components in the hidden representations will naturally decay over context and layers. This implies that LLMs performing ICL should be inherently robust to high-frequency noise. Since natural signals are typically dominated by low-frequency structure (Field, 1987), this suggests that ICL should be able to tolerate and even correct a moderate amount of errors in the input.

To test this prediction, we conduct an additional experiment under a noisy data-generating process. Specifically, during the random walk over the graph $\mathcal{G}$, we inject noise by allowing the sequence to transition to a uniformly random token in $[c]$ with $1\%$ probability at each step, rather than to a graph neighbor. This corruption can be viewed as temporarily replacing the original graph with a complete graph, which introduces purely high-frequency components into the sequence.

In Figure 4, we measure the number of non-neighbor transitions (i.e. token pairs that do not correspond to valid edges in $\mathcal{G}$) within a sliding window of the last 500 tokens. We compare this quantity in both the input sequence and the output predicted by the ICL model. While the input maintains a constant error rate due to the injected corruption, the ICL output gradually eliminates these errors. Once the context becomes sufficiently long, the model consistently produces transitions that respect the original graph structure, effectively achieving $100\%$ accuracy despite the noisy input.

This result provides further evidence that ICL dynamics favor low-frequency structure and can suppress high-frequency perturbations. This result also explains previous observations that ICL can implicitly denoise input data (Cheng et al., 2025; Alazraki et al., 2025).

## 6 Summary

In this paper, we investigate the dynamics of hidden representations across both context length and depth (layers) in a pre-trained Transformer. Under reasonable assumptions on the attention maps and the DGP, we formally prove a double convergence phenomenon: hidden representations converge both as the input context grows and as the model depth increases. The limit representations exhibit a low-frequency bias, which accounts for several previously observed phenomena in Park et al. (2024), as well as the widely observed robustness of ICL under high-frequency noise (Cheng et al., 2025; Alazraki et al., 2025).

In addition to the main theoretical results, we provide general techniques in the appendix that relax the dependence on any specific DGP. We hypothesize that the low-frequency bias is a universal property of pre-trained Transformers given data generated by Markov processes. That being said, the most detailed results in this paper still focus on the specific DGP proposed in Park et al. (2024). Generalizing our analysis to broader data distributions remains an open and important future direction.

Finally, this work suggests a path towards an end-to-end theoretical understanding of low-frequency bias in LLMs. A key next step is to analyze the pre-training process itself, and show how the structural assumptions we impose on attention maps naturally emerge from gradient-based learning during training.

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

## A BACKGROUND AND RELATED WORK

In this section, we briefly review the current empirical and theoretical understanding of the mechanisms underlying ICL and the structure of hidden representations in Transformers. We highlight the limitations and challenges faced by existing studies and explain how the perspective adopted in this paper offers a potential path forward.

**Empirical Explorations of the Mechanisms of In-Context Learning**  Other than (Park et al., 2024), there is also a lot of empirical work trying to explore the mechanisms of ICL through empirical studies (Min et al., 2022b; Pan et al., 2023; Zhao et al., 2024; Wurgaft et al., 2025), typically by probing model behavior through controlled prompting setups, task decomposition, or architectural comparisons. However, these studies primarily analyze input-output behaviors or high-level functional properties of models and therefore offer only indirect evidence regarding the internal computational structure. One recent work (Singh Lubana et al., 2025) have characterized the hidden representation in ICL as a decomposition of so-called predictable and residual components, where the predictable component dominate the representation, and it shows similar ICLR behavior in real text, further supporting the findings of (Park et al., 2024).

**Towards a Theoretical Understanding of the Mechanisms of In-Context Learning**  Understanding the mechanisms behind ICL has become a central topic in deep learning research. However, progress has been limited by the highly complex architecture and learning dynamics of neural networks. Existing theoretical approaches can be broadly categorized into two lines of work: 1) *Existential results*: This line of work constructs specific Transformer implementations that implement certain in-context algorithms, hence proving the existence of Transformers capable of performing in-context learning (Akyürek et al., 2022b; Dai et al., 2022; Von Oswald et al., 2023; Li et al., 2025). 2) *Learning dynamics or loss landscape of simplified models*: Another line of work studies learning dynamics or the structure of the loss landscape in simplified Transformer settings, typically with a small number of layers or restricted architectures. These studies show that the models can converge to configurations that exhibit in-context learning behaviors. However, due to the complexity of Transformer training dynamics, these analyses are usually restricted to one-layer Transformers (Lu et al., 2024; Huang et al., 2023), linearized models (Ahn et al., 2023), or two-layer models with controlled training setups (Wang et al., 2024).

**Structure of Hidden Representations**  Another related line of theoretical research investigates how hidden representations evolve across layers during inference (this topic is sometimes referred to as "inference dynamics") (Ramsauer et al., 2020; Yang et al., 2022; Yu et al., 2023; Geshkovski et al., 2023; Tomihari & Karakida, 2025; Hu et al., 2025). Despite their insights, these studies also typically require simplifying or modifying the Transformer architecture due to the non-linear and heterogeneous nature of Transformers. Yang et al. (2022) outlined four major challenges in analyzing inference dynamics, many of which remain unsolved.

**Structure of Attention Maps**  Olsson et al. (2022), identified a specific attention mechanism known as *induction heads*. Generally speaking, they are attention heads that implement a form of token copying: they identify a previous occurrence of the current token and attend to the token that follows it. Formally, let the input tokens be $\{x_k\}_{k=1}^n$, generated by a *Markov process*, and the attention layer be defined as $\boldsymbol{u}_k = \sum_{j=1}^k a_{k,j} \boldsymbol{v}_k$, where $\{\boldsymbol{v}_k\}_{k=1}^n$ and $\{\boldsymbol{u}_k\}_{k=1}^n$ are input and output representations respectively, and $\{a_{k,j}\}_{k,j \in [n]}$ denotes the attention weights, then the induction heads can be defined as attention maps satisfying the following condition:

$$a_{k,j} > 0 \implies x_j \in \mathcal{N}(x_k), \tag{11}$$

where $\mathcal{N}(x)$ is the set of all possible next tokens of $x$.

A major challenge in existing methods when analyzing inference dynamics arises from the interactive and heterogeneous structure of Transformer models: the attention map depends on the hidden representations, which in turn evolve through both self-attention and feedforward layers. This bidirectional dependency makes theoretical analysis extremely difficult, as noted by Yang et al. (2022). In this paper, to overcome this difficulty, we adopt a more pragmatic approach: Rather than attempting to derive the structure of attention maps from first principles, we posit a structured form for the attention map, that is strong enough to enable meaningful theoretical results, yet general enough to be empirically validated and extendable to broader settings. Our goal is not to explain why attention maps take this form, but to demonstrate that, if they do, they can give rise to the

observed structure in hidden representations, and empirically verify the validity of these assumptions in practice.

## B  PROOF THEORETICAL RESULTS W.R.T. CONTEXT-WISE CONVERGENCE

In this section, we prove Theorem 1. The proof start by identifying a high-probability event in the random walk sequence that ensures it is "regular" enough.

### B.1  EVENTS IN A RANDOM WALK SEQUENCE

**Theorem 3** (Theorem 1 in (Fan et al., 2021)). *Given graph $\mathcal{G} = (\mathcal{V}, \mathcal{E})$ with stationary distribution $\boldsymbol{\pi}$, there exists a constant $C > 0$ that satisfies the following statement. Let $\{x_i\}_{i=1}^{\infty} \in \mathcal{V}^{\mathbb{N}}$ be a random walk sequence on $\mathcal{G}$ starting from the stationary distribution, and $\{f_i\}_{i=1}^{\infty}$ be a sequence of bounded functions satisfying $f_i(\mathcal{V}) \subseteq [-\alpha_i, \alpha_i]$, then for any $k \in \mathbb{N}$ and $\epsilon > 0$,*

$$\mathbb{P}\left\{\left|\sum_{i=1}^{k} f_i(x_i) - \sum_{i=1}^{k} \boldsymbol{\pi}(f_i)\right| > \epsilon\right\} \leq 2\exp\left(-\frac{C\epsilon^2}{k\sum_{k=1}^{k}\alpha_i^2}\right), \tag{12}$$

*where $\boldsymbol{\pi}(f_i) = \sum_{x \in \mathcal{V}} \pi_x f_i(x)$ is the expectation of $f_i$ under the distribution defined by $\boldsymbol{\pi}$.*

Notice that in theorem 3, we view the spectral property of the graph as constant and absorb it into $C$. Below is a direct corollary of Theorem 3.

**Corollary 4.** *There exists constants $C, C'$ (that probably depends on $\mathcal{G}$) such that the following inequality holds.*

$$\forall S \subseteq [c], \mathbb{P}\left\{\left|\frac{\sum_{y \in S} F_{y,k}}{k} - \sum_{y \in S} \pi_y\right| > \epsilon\right\} \leq 2\exp\left(-C\epsilon^2 k\right), \tag{13}$$

*in other words, with probability at least 0.999, we have*

$$\forall k \in [n], \forall S \subseteq [c], \left|\frac{\sum_{y \in S} F_{y,k}}{k} - \sum_{y \in S} \pi_y\right| \leq \frac{C' \log(n)}{\sqrt{k}}. \tag{14}$$

Notice that we assumed $n > 10c$, therefore the following statement is also a direct corollary of Theorem 3.

**Corollary 5.** *The following statement holds with probability at least 0.999: for any $x \in [c]$, $F_{x,n} \geq F_{x,\lceil n/2 \rceil} + 1$.*

### B.2  PROOF OF THEOREM 1

Now we prove Theorem 1. We start with a lemma that is easy to verify.

**Lemma 6.** *If $a, b, r, s > 0$ satisfies $|a - r| \leq \epsilon$ and $|b - s| \leq \epsilon$ and $\epsilon \leq s/2$, then $\left|\frac{a}{b} - \frac{r}{s}\right| \leq 2\epsilon \frac{r+s}{s^2}$.*

*Proof.* Let $\{\boldsymbol{u}_k\}_{k=1}^{n} = \boldsymbol{A}\mathcal{V}$. Let $\boldsymbol{z}'_x = \frac{\sum_{y \in f(x)} \pi_y \boldsymbol{z}_y}{\sum_{y \in f(x)} \pi_y}$.

For $k \leq c$, we have

$$\|\boldsymbol{u}_k - \boldsymbol{z}'_k\| \leq \sum_{j=1}^{k} a_{k,j} \|\boldsymbol{v}_j\| + \|\boldsymbol{z}'_k\| \leq \sum_{j=1}^{k} \psi(\gamma + N) + N \leq \frac{\sqrt{c}(c+1)(\psi\gamma + \psi N)}{\sqrt{k}}. \tag{15}$$

Therefore, the error can be absorbed into the $C\psi(\gamma + N)$ terms since $C$ is allowed to be dependent on $c$. Below we only consider $k > c$. Moreover, if $f(x_k) = \emptyset$, then it is obvious that $\boldsymbol{u}_k = \boldsymbol{0} = \boldsymbol{z}'_{x_k}$. Therefore, in the following we only consider the case where $f(x) \neq \emptyset$.

Let $R_k = \sum_{y \in f(x)} F_{y,k}$. Let $\tilde{z}_x^{(k)} = \sum_{y \in f(x)} \frac{F_{y,k} z_y}{R_k}$. We have

$$
\left\| u_k - \tilde{z}_x^{(k)} \right\| = \left\| \sum_{\substack{y \in f(x)}} \left( \sum_{\substack{j \in [k] \\ x_j = y}} a_{k,j} v_j - \frac{z_y F_{y,j}}{R_k} \right) \right\| \tag{16}
$$

$$
= \left\| \sum_{\substack{y \in f(x)}} \sum_{\substack{j \in [k] \\ x_j = y}} \left( a_{k,j} v_j - \frac{z_{x_j}}{R_k} \right) \right\| \tag{17}
$$

$$
= \left\| \sum_{\substack{y \in f(x)}} \left( \sum_{\substack{j \in [k] \\ x_j = y}} \left( a_{k,j} v_j - z_{x_j} \right) \right) + \sum_{\substack{y \in f(x)}} z_y \sum_{\substack{j \in [k] \\ x_j = k}} \left( a_{k,j} - \frac{1}{R_k} \right) \right\| \tag{18}
$$

$$
\leq \sum_{\substack{j \in [k] \\ x_j \in f(x)}} a_{k,j} \left\| v_j - z_{x_j} \right\| + N \sum_{\substack{y \in f(x)}} \left| \frac{F_{y,k}}{R_k} - \sum_{\substack{j \in [k] \\ x_j = y}} a_{k,j} \right| \tag{19}
$$

$$
\overset{\text{(i)}}{\leq} \gamma \left( \sum_{\substack{j \in [k] \\ x_j \in f(x)}} \frac{a_{k,j}}{\sqrt{j}} \right) + \frac{N c \psi}{\sqrt{k}} \tag{20}
$$

$$
\leq \gamma \left( \sum_{j=1}^{k} \frac{a_{k,j}}{\sqrt{j}} \right) + \frac{N c \psi}{\sqrt{k}}, \tag{21}
$$

where in (i) we use the condition that $\mathcal{V}$ is a stable sequence converging to $\mathcal{Z}$ and that $\mathcal{A}$ reflects $f$.

Now, define $S_{k,j} = \sum_{i=1}^{j} a_{k,j}$ and $S_{k,0} = 0$. Since $A$ is a balanced attention map with parameter $\psi$, we have $S_j \leq \frac{\psi j}{k}$. Notice that $a_{k,j} = S_{k,j} - S_{k,j-1}$. Thus we have

$$
\sum_{j=1}^{k} \frac{a_{k,j}}{\sqrt{j}} = \sum_{j=1}^{k} \frac{1}{\sqrt{j}} \left( S_{k,j} - S_{k,j-1} \right) \tag{22}
$$

$$
= \frac{S_{k,k}}{\sqrt{k}} + \sum_{j=0}^{k-1} S_{k,j} \left( \frac{1}{\sqrt{j}} - \frac{1}{\sqrt{j+1}} \right) \tag{23}
$$

$$
\leq \frac{\psi}{\sqrt{k}} + \frac{\psi}{k} \sum_{j=1}^{k-1} j \left( \frac{1}{\sqrt{j}} - \frac{1}{\sqrt{j+1}} \right) \tag{24}
$$

$$
= \frac{\psi}{\sqrt{k}} + \frac{\psi}{k} \sum_{j=1}^{k-1} \frac{1}{\sqrt{j}} - \frac{\psi(k-1)}{k\sqrt{k}} \tag{25}
$$

$$
= \frac{\psi}{k} \sum_{j=1}^{k} \frac{1}{\sqrt{j}}. \tag{26}
$$

Notice that,

$$
\sum_{j=1}^{k} \frac{1}{\sqrt{j}} = 1 + \sum_{j=2}^{k} \int_{j-1}^{j} \frac{1}{\sqrt{j}} \mathrm{d}x \leq 1 + \sum_{j=2}^{k} \int_{j-1}^{j} \frac{1}{\sqrt{x}} \mathrm{d}x = 1 + \int_{1}^{k} \frac{1}{\sqrt{x}} \mathrm{d}x = 2\sqrt{k}. \tag{27}
$$

Subtracting eq. (27) into the argument above, we obtain

$$
\left\| u_k - \tilde{z}_x^{(k)} \right\| \leq \frac{2\psi\gamma + N c \psi}{\sqrt{k}}. \tag{28}
$$

Moreover,

$$\left\| \widetilde{\boldsymbol{z}}_x^{(k)} - \boldsymbol{z}_x' \right\| = \left\| \sum_{y \in f(x)} \left( \frac{F_{y,k}}{R} - \frac{\pi_y}{\sum_{y' \in f(x)} \pi_{y'}} \right) \boldsymbol{z}_y \right\| \tag{29}$$

$$\leq N \sum_{y \in [c]} \left| \frac{F_{y,k}/k}{R/k} - \frac{\pi_y}{\sum_{y' \in f(x)} \pi_{y'}} \right|. \tag{30}$$

Define $a_{k,y} = F_{y,k}/k$, $b_{k,y} = R/k$, $r_y = \pi_y$ and $s_y = \sum_{y' \in f(x)} \pi_{y'}$. From Corollary 4 we have there exists a constant number $C > 0$ that only depends on $\mathcal{G}$, and an event whose probability is at least $0.999$, such that for all $k \in [n]$ and $y \in [c]$ we have $\max\{|a_{y,k} - r_y|, |b_{k,y} - s_y|\} \leq \frac{C \log n}{\sqrt{k}}$ (notice that this event is only related to the random walk, and does not depend on the specific values of $\mathcal{V}$, $\boldsymbol{A}$, etc.).

Let $\rho = \min_{y \in [c]} \pi_y \in (0,1)$. Let $C' = 2C/\rho > C$ that also only depends on $\mathcal{G}$. If $k \geq \frac{4C^2 (\log n)^2}{\rho^2}$, then $\frac{C \log n}{\sqrt{k}} \leq \frac{\rho}{2} \leq \frac{s_y}{2}$, thus from Lemma 6 we have $\left| \frac{a_{y,k}}{r_y} - \frac{b_{y,k}}{s_y} \right| \leq \frac{C \log n}{\sqrt{k}} \leq \frac{C' \log n}{\sqrt{k}}$. On the other hand, if $k < \frac{4C^2 (\log n)^2}{\rho^2}$, we have $\frac{C' \log n}{\sqrt{k}} \geq 1 \geq \left| \frac{a_{y,k}}{r_y} - \frac{b_{y,k}}{s_y} \right|$. Therefore, we conclude that, in an event at happens with probability at least $0.999$, for all $k \in [n]$ and $y \in [c]$ we have $\left| \frac{F_{y,k}}{R} - \frac{\pi_y}{\sum_{y' \in f(x)} \pi_{y'}} \right| \leq \frac{C' \log n}{\sqrt{k}}$.

Combining the above arguments, we conclude that with probability at least $0.999$, it holds that for all $k$,

$$\left\| \boldsymbol{u}_k - \boldsymbol{z}_{x_k}' \right\| \leq \frac{Nc\psi + 2\psi\gamma + C'c \log n}{\sqrt{k}}, \tag{31}$$

which is the desired conclusion. $\qquad\square$

## C  PROOF OF THEORETICAL RESULTS W.R.T. LAYER-WISE CONVERGENCE

We first prove that under the specific conditions given in Section 4, how does the limiting representations evolves.

**Lemma 7.** *Suppose $\mathcal{V} = \{\boldsymbol{v}_k\}_{k=1}^n \in \left( \mathbb{R}^d \right)^n$ is a stable sequence converging to $\mathcal{Z} = \{\boldsymbol{z}_x\}_{x \in [c]}$ with parameter $\gamma$, and $\mathcal{V}' = \{\boldsymbol{v}_k'\}_{k=1}^n \in \left( \mathbb{R}^d \right)^n$ is a stable sequence converging to $\mathcal{Z}' = \{\boldsymbol{z}_x'\}_{x \in [c]}$ with parameter $\gamma'$. Moreover, suppose $T, G : \mathbb{R}^d \to \mathbb{R}^d$ are Lipschitz continuous functions with Lipschitz constants $L_T$, $L_G$ respectively. Then, we have $\{T\boldsymbol{v}_k + G\boldsymbol{v}_k'\}$ is a stable sequence converging to $\{T\boldsymbol{z}_x + G\boldsymbol{z}_x'\}_{x \in \mathcal{V}}$ with parameter $L_T \gamma + L_G \gamma'$.*

*Proof.* Only need to notice that for any $k \in [n]$,

$$\left\| (T\boldsymbol{v}_k + G\boldsymbol{v}_k') - (T\boldsymbol{z}_{x_k} + G\boldsymbol{z}_{x_k}') \right\| \leq L_T \left\| \boldsymbol{v}_k - \boldsymbol{z}_{x_k} \right\| + L_G \left\| \boldsymbol{v}_k' - \boldsymbol{z}_{x_k}' \right\| \leq \frac{L_T \gamma + L_G \gamma'}{\sqrt{k}}. \tag{32}$$
$$\qquad\square$$

**Lemma 8.** *There exists a scalar number $C > 0$ that possibly depends on the graph $\mathcal{G}$, and an event with probability at least $0.999$, such that the following statement holds. For any layer $\ell$, if $\mathcal{V} = \{\boldsymbol{v}_k\}_{k=1}^n$ is a stable sequence converging to $\mathcal{Z} = \{\boldsymbol{z}_x\}_{x \in [c]}$ with parameter $\gamma$, and let $\boldsymbol{A}^{(\ell)}$ be defined as in eq. (6), then $\mathcal{U} = \boldsymbol{A}^{(\ell)} \mathcal{V}$ is a stable sequence converging to $\boldsymbol{Z}' = \{\boldsymbol{z}_x'\}_{x \in [c]}$, where*

$$\boldsymbol{z}_x' = \rho_A^{(\ell)} \boldsymbol{z}_x + \rho_B^{(\ell)} \sum_{y \in [c]} \frac{w_{x,y}}{d_x} \boldsymbol{z}_y + \rho_O^{(\ell)} \sum_{y \in [c]} \pi_{\mathcal{G}}(y) \boldsymbol{z}_y + \rho_T^{(\ell)} \boldsymbol{v}_1, \tag{33}$$

*with parameter*

$$\kappa = C(\gamma + N) \left( \sum_{\tau \in \{A,B,O\}} \rho_\tau^{(\ell)} \psi_\tau^{(\ell)} \right) + C \log n \sum_{\tau \in \{A,B,O\}} \rho_\tau^{(\ell)}, \tag{34}$$

*where $N = \max_{y \in [c]} \|\boldsymbol{z}_y\|$.*

*Proof.* Let $\widehat{\boldsymbol{A}}^{(\ell)} = \boldsymbol{A}^{(\ell)} - \rho_T^{(\ell)} \boldsymbol{A}^{(T)}$, and let $\widehat{\mathcal{U}} = \left( \widehat{\boldsymbol{A}}^{(\ell)} \mathcal{V}; \widehat{\mathcal{Z}}' \right)$, where $\widehat{\mathcal{Z}}' = \{\widehat{\boldsymbol{z}}'_x\}_{x \in [c]}$ is defined as

$$\widehat{\boldsymbol{z}}'_x = \boldsymbol{z}'_x - \rho_T^{(\ell)} \boldsymbol{v}_1. \tag{35}$$

For a token $x \in [c]$, let $\mathcal{N}(x)$ be the set of all neighbors of $x$. Notice that for any $y \in \mathcal{N}(x)$, we have

$$\frac{\pi_y}{\sum_{y' \in \mathcal{N}(x)} \pi_{y'}} = \frac{\pi_x \pi_y}{\pi_x \sum_{y' \in [c]} \widetilde{w}_{x,y'} \pi_{y'}} = \frac{w_{x,y}}{d_x}. \tag{36}$$

Therefore, from Lemma 7 and Theorem 1, we have $\widehat{\mathcal{U}}$ converges to $\widehat{\mathcal{Z}}'$ with parameter $\kappa$.

Since for $k > c$, $\boldsymbol{u}_k = \widehat{\boldsymbol{u}}_k + \rho_T^{(\ell)} \sum_{j=1}^{k} a_{k,j}^{(T)} \boldsymbol{v}_j = \widehat{\boldsymbol{u}}_k + \rho_T^{(\ell)} \boldsymbol{v}_1$, and $\boldsymbol{z}'_{x_k} = \widehat{\boldsymbol{z}}'_x + \rho_T^{(\ell)} \boldsymbol{v}_1$, we have

$$\|\boldsymbol{u}_k - \boldsymbol{z}'_k\| = \|\widehat{\boldsymbol{u}}_k - \widehat{\boldsymbol{z}}'_k\| \le \frac{\kappa}{\sqrt{k}}, \tag{37}$$

we have $\mathcal{U}$ is also a stable sequence converging to $\mathcal{Z}'$ with parameter $\kappa$. $\qquad\square$

### C.1 EVOLUTION OF THE LIMITING REPRESENTATION

From this sub-section, we focus on the evolution of the limiting representation across layers and show where do they converge.

**Lemma 9.** *Let $\rho_A, \rho_B, \rho_O, \rho_T > 0$. Let $\boldsymbol{D} = \mathrm{diag}\,(\boldsymbol{W}\boldsymbol{1})$ is the degree matrix. Let $\boldsymbol{z} \in \mathbb{R}^c$ be a vector, and let $\boldsymbol{z}'$ be defined as*

$$\boldsymbol{z}' = \rho_A \boldsymbol{z} + \rho_B \boldsymbol{D}^{-1} \boldsymbol{W} \boldsymbol{z} + \rho_O \langle \boldsymbol{\alpha}, \boldsymbol{z} \rangle \boldsymbol{1} + \rho_T \boldsymbol{1}. \tag{38}$$

*Then, for any $\boldsymbol{U}^\top$ be a projection on to a subspace orthogonal to $\boldsymbol{D}^{1/2}\boldsymbol{1}$, we have*

$$\boldsymbol{U}^\top \boldsymbol{D}^{1/2} \boldsymbol{z}' = \boldsymbol{U}^\top \boldsymbol{M} \boldsymbol{D}^{1/2} \boldsymbol{z}, \tag{39}$$

*where $\boldsymbol{M} = \rho_A \boldsymbol{I} + \rho_B \boldsymbol{D}^{-1/2} \boldsymbol{W} \boldsymbol{D}^{-1/2}$.*

*Proof.* Let $\boldsymbol{E} = \boldsymbol{D}^{1/2} \boldsymbol{1} \boldsymbol{\alpha}^\top \boldsymbol{D}^{-1/2}$. We have

$$\boldsymbol{z}' = \left( \rho_A \boldsymbol{I} + \rho_B \boldsymbol{D}^{-1} \boldsymbol{W} + \rho_O \boldsymbol{1} \boldsymbol{\alpha}^\top \right) \boldsymbol{z} + \rho_T \boldsymbol{1} \tag{40}$$

$$= \boldsymbol{D}^{-1/2} \left( \rho_A \boldsymbol{I} + \rho_B \boldsymbol{D}^{-1} \boldsymbol{W} + \rho_O \boldsymbol{1} \boldsymbol{\alpha}^\top \right) \boldsymbol{D}^{1/2} \boldsymbol{z} + \rho_T \boldsymbol{1}. \tag{41}$$

$$= \boldsymbol{D}^{-1/2} \left( \boldsymbol{M} + \rho_O \boldsymbol{E} \right) \boldsymbol{D}^{1/2} \boldsymbol{z} + \rho_T \boldsymbol{1}. \tag{42}$$

Let $\widetilde{\boldsymbol{z}}' = \boldsymbol{D}^{1/2} \boldsymbol{z}'$, $\widetilde{\boldsymbol{z}} = \boldsymbol{D}^{1/2} \boldsymbol{z}$, and $\widetilde{\boldsymbol{1}} = \boldsymbol{D}^{1/2} \boldsymbol{1}$. Thus, we have

$$\widetilde{\boldsymbol{z}}' = \boldsymbol{M} \widetilde{\boldsymbol{z}} + \rho_O \boldsymbol{E} \widetilde{\boldsymbol{z}} + \rho_T \widetilde{\boldsymbol{1}} \tag{43}$$

Notice that, since $\boldsymbol{E}$ is a rank-1 matrix, its image space is $\mathrm{span}\,\widetilde{\boldsymbol{1}}$: for any vector $\boldsymbol{x} \in \mathbb{R}^c$,

$$\boldsymbol{E}\boldsymbol{x} = \boldsymbol{D}^{1/2} \boldsymbol{1} \boldsymbol{\alpha}^\top \boldsymbol{D}^{-1/2} \boldsymbol{x} = \left\langle \boldsymbol{\alpha}, \boldsymbol{D}^{-1/2} \boldsymbol{x} \right\rangle \boldsymbol{D}^{1/2} \boldsymbol{1} = \left\langle \boldsymbol{\alpha}, \boldsymbol{D}^{-1/2} \boldsymbol{x} \right\rangle \widetilde{\boldsymbol{1}}. \tag{44}$$

Therefore, we have

$$\boldsymbol{U}^\top \widetilde{\boldsymbol{z}}' = \boldsymbol{U}^\top \boldsymbol{M} \widetilde{\boldsymbol{z}} + \rho_O \boldsymbol{U}^\top \left( \boldsymbol{E} \widetilde{\boldsymbol{z}} \right) + \rho_T \boldsymbol{U}^\top \widetilde{\boldsymbol{1}} \tag{45}$$

$$= \boldsymbol{U}^\top \boldsymbol{M} \widetilde{\boldsymbol{z}} + \left( \rho_O \left\langle \boldsymbol{\alpha}, \boldsymbol{D}^{-1/2} \widetilde{\boldsymbol{z}} \right\rangle + \rho_T \right) \boldsymbol{U}^\top \widetilde{\boldsymbol{1}} \tag{46}$$

$$= \boldsymbol{U}^\top \boldsymbol{M} \widetilde{\boldsymbol{z}}. \tag{47}$$

$\qquad\square$

**Corollary 10.** *Let $\rho_A, \rho_B, \rho_O, \rho_T > 0$. Let $\mathcal{Z} = \{\boldsymbol{z}_x\}_{x \in [c]} \in \left( \mathbb{R}^d \right)^c$ be a sequence, and define sequence $\mathcal{Z}' = \{\boldsymbol{z}'_x\}_{x \in [c]}$ as follows:*

$$\boldsymbol{z}'_x = \rho_A \boldsymbol{z}_x + \frac{\rho_B}{d_x} \sum_{y \in [c]} w_{x,y} \boldsymbol{z}_y + \rho_O \sum_{y \in [c]} \alpha_y \boldsymbol{z}_y + \rho_T \boldsymbol{v}_1. \tag{48}$$

*Let $\boldsymbol{Z} = \operatorname{mat} \mathcal{Z} \in \mathbb{R}^{d \times c}$ and $\boldsymbol{Z}' = \operatorname{mat} \mathcal{Z}' \in \mathbb{R}^{d \times c}$. Let $\boldsymbol{M} = \rho_A \boldsymbol{I} + \rho_B \boldsymbol{D}^{-1/2} \boldsymbol{W} \boldsymbol{D}^{-1/2}$, and $\{\lambda_k\}_{k=1}^n$ be its eigenvalues, arranged in a non-increasing order of absolute values. Let the eigen-decomposition of $\boldsymbol{M}$ be*

$$\boldsymbol{M} = \begin{bmatrix} \boldsymbol{f} & \boldsymbol{X} & \boldsymbol{Y} \end{bmatrix} \begin{bmatrix} \lambda_1 & & \\ & \boldsymbol{\Lambda} & \\ & & \boldsymbol{\Lambda}' \end{bmatrix} \begin{bmatrix} \boldsymbol{f}^\top \\ \boldsymbol{X}^\top \\ \boldsymbol{Y}^\top, \end{bmatrix} \tag{49}$$

*where $\boldsymbol{\Lambda} = \{\lambda_k\}_{k=2}^q$ and $\boldsymbol{\Lambda}' = \{\lambda_k\}_{k=q+1}^c$. Then, we have*

$$\frac{\left\| \boldsymbol{Z}' \boldsymbol{D}^{1/2} \boldsymbol{X} \right\|}{\left\| \boldsymbol{Z} \boldsymbol{D}^{1/2} \boldsymbol{X} \right\|} \geq \delta_M \frac{\left\| \boldsymbol{Z}' \boldsymbol{D}^{1/2} \boldsymbol{Y} \right\|}{\left\| \boldsymbol{Z} \boldsymbol{D}^{1/2} \boldsymbol{Y} \right\|} \tag{50}$$

*Proof.* From Lemma 9, we have

$$\boldsymbol{Z}' \boldsymbol{D}^{1/2} \boldsymbol{X} = \boldsymbol{Z} \boldsymbol{D}^{1/2} \boldsymbol{M} \boldsymbol{X} = \boldsymbol{Z} \boldsymbol{D}^{1/2} \boldsymbol{X} \boldsymbol{\Lambda}, \tag{51}$$

therefore

$$\left\| \boldsymbol{Z}' \boldsymbol{D}^{1/2} \boldsymbol{X} \right\|_F = \left\| \boldsymbol{Z} \boldsymbol{D}^{1/2} \boldsymbol{X} \boldsymbol{\Lambda} \right\| \geq \left\| \boldsymbol{Z} \boldsymbol{D}^{1/2} \boldsymbol{X} \right\|_F \|\boldsymbol{\Lambda}\| \geq |\lambda_q| \left\| \boldsymbol{Z} \boldsymbol{D}^{1/2} \boldsymbol{X} \right\|_F. \tag{52}$$

Similarly, we have

$$\left\| \boldsymbol{Z}' \boldsymbol{D}^{1/2} \boldsymbol{Y} \right\|_F \leq |\lambda_{q+1}| \left\| \boldsymbol{Z} \boldsymbol{D}^{1/2} \boldsymbol{Y} \right\|_F. \tag{53}$$

The proposition directly follows. $\qquad \square$

**Corollary 11.** *Under the same condition as in Corollary 10, let $\sigma : \mathbb{R}^d \to \mathbb{R}^d$ be a great mapping w.r.t. $(\gamma_1, \gamma_2, \{\boldsymbol{Z}'\}, \boldsymbol{X})$ and $(\gamma_1', \gamma_2', \{\boldsymbol{Z}'\}, \boldsymbol{Y})$. Then, we have*

$$\frac{\left\| \sigma\left(\boldsymbol{Z}'\right) \boldsymbol{D}^{1/2} \boldsymbol{X} \right\|}{\left\| \boldsymbol{Z} \boldsymbol{D}^{1/2} \boldsymbol{X} \right\|} \geq \delta_M \frac{\gamma_1}{\gamma_2'} \frac{\left\| \sigma\left(\boldsymbol{Z}'\right) \boldsymbol{D}^{1/2} \boldsymbol{Y} \right\|}{\left\| \boldsymbol{Z} \boldsymbol{D}^{1/2} \boldsymbol{Y} \right\|}. \tag{54}$$

*Proof.* Only need to repeat the proof of Corollary 10 and use the definition of great mappings. Notice that

$$\left\| \sigma\left(\boldsymbol{Z}'\right) \boldsymbol{D}^{1/2} \boldsymbol{X} \right\|_F \geq \gamma_1 \left\| \boldsymbol{Z}' \boldsymbol{D}^{1/2} \boldsymbol{X} \right\|_F \geq \gamma_1 |\lambda_q| \left\| \boldsymbol{Z}' \boldsymbol{D}^{1/2} \boldsymbol{X} \right\|_F. \tag{55}$$

Similarly,

$$\left\| \sigma\left(\boldsymbol{Z}'\right) \boldsymbol{D}^{1/2} \boldsymbol{Y} \right\|_F \leq \gamma_2' \left\| \boldsymbol{Z}' \boldsymbol{D}^{1/2} \boldsymbol{Y} \right\|_F \leq \gamma_2' |\lambda_{q+1}| \left\| \boldsymbol{Z}' \boldsymbol{D}^{1/2} \boldsymbol{Y} \right\|_F. \tag{56}$$

The proposition directly follows. $\qquad \square$

## C.2 Proof of Theorem 2

Let $E_1$ be an event with probability at least $0.999$ defined in Lemma 8. Let $E_2 = \{\forall x \in [c], n_{[x]} > n/2\}$, Corollary 5 shows that $E_2$ also holds with probability at least $0.999$. In the following, we condition on the event $E_1 \cap E_2$, which holds with probability at least $0.99$.

Let $\mathcal{Z}^{(\ell)} = \left\{ \boldsymbol{z}_x^{(\ell)} \right\}_{x=1}^c \in \left(\mathbb{R}^d\right)^c$ be defined as follows: $\boldsymbol{z}_x^{(1)} = \boldsymbol{b}_x$, and

$$\boldsymbol{z}_x^{(\ell+1)} = \rho_A^{(\ell)} \boldsymbol{z}_x^{(\ell)} + \frac{\rho_B^{(\ell)}}{d_x} \sum_{y \in [c]} w_{x,y} \boldsymbol{z}_y^{(\ell)} + \rho_O \sum_{y \in [c]} \alpha_y \boldsymbol{z}_y^{(\ell)} + \rho_T^{(\ell)} \boldsymbol{v}_1^{(\ell)}. \tag{57}$$

for any $\ell \in [L-1]$, and $\boldsymbol{z}_x'^{(\ell)} = \sigma^{(\ell)}\left(\boldsymbol{z}_k^{(\ell)}\right)$. From the definition, it is obvious that $\left\{ \boldsymbol{v}_k^{(1)} \right\}_{k=1}^n$ converges to $\mathcal{Z}^{(1)}$ with parameter $0$.

- We first fix an $L \in \mathbb{N}$ and analyze the context-wise convergence. First consider an arbitrary $x \in [c]$. Using Lemma 8 with an induction, it is not hard to prove that for each $\ell \in L$ there exists a $\gamma^{(\ell)} = \mathrm{poly} \log n$ (since we are going to take limit w.r.t. $n$, we view all other values independent of $n$ as constants; notice that $\ell$ is a fixed index here), such that $\left\{\boldsymbol{v}_k^{(\ell)}\right\}_{k=1}^n$ is a stable sequence converging to $\left\{\boldsymbol{z}_y'^{(\ell)}\right\}_{y \in [c]}$ with parameter $\gamma^{(\ell)}$. Therefore, we have

$$\left\|\boldsymbol{v}_{n_{[x]}}^{(\ell)} - \sigma\left(\boldsymbol{z}_x^{(\ell)}\right)\right\| \leq \frac{\gamma^{(\ell)}}{\sqrt{n_{[x]}}} < \frac{\sqrt{2}\,\mathrm{poly}\log n}{\sqrt{n}}. \tag{58}$$

Therefore, taking $\ell = L$ and sending $n \to \infty$, we have

$$\lim_{n \to \infty} \left\|\boldsymbol{v}_{n_{[x]}}^{(L)} - \boldsymbol{z}_x'^{(L)}\right\| = 0, \tag{59}$$

which is equivalent to

$$\lim_{n \to \infty} \boldsymbol{v}_{n_{[x]}}^{(L)} = \boldsymbol{z}_x'^{(\ell)}. \tag{60}$$

Notice that this holds for any $x \in [c]$. Therefore, let $\boldsymbol{Z}'^{(\ell)} = \mathrm{mat}\left\{\boldsymbol{z}_x'^{(\ell)}\right\}_{x \in [c]} \in \mathbb{R}^{d \times c}$, we have when $\left\|\boldsymbol{V}_n'^{(L)}\boldsymbol{D}^{1/2}\boldsymbol{X}\right\| > 0$, we have

$$\lim_{n \to \infty} \frac{\left\|\boldsymbol{V}_n^{(L)}\boldsymbol{D}^{1/2}\boldsymbol{Y}\right\|}{\left\|\boldsymbol{V}_n^{(L)}\boldsymbol{D}^{1/2}\boldsymbol{X}\right\|} = \frac{\left\|\boldsymbol{Z}'^{(L)}\boldsymbol{D}^{1/2}\boldsymbol{Y}\right\|}{\left\|\boldsymbol{Z}'^{(L)}\boldsymbol{D}^{1/2}\boldsymbol{X}\right\|}. \tag{61}$$

- Next, we consider the layer-wise evolution. Corollary 11 shows that for any $\ell \in [L]$,

$$\frac{\left\|\boldsymbol{Z}'^{(\ell)}\boldsymbol{D}^{1/2}\boldsymbol{Y}\right\|}{\left\|\boldsymbol{Z}'^{(\ell)}\boldsymbol{D}^{1/2}\boldsymbol{X}\right\|} \leq \left[\delta_q\left(\rho_A\boldsymbol{I} + \rho_B\boldsymbol{M}\right)\right]^{-1} \frac{\gamma_2'^{(\ell)}}{\gamma_1^{(\ell)}} \frac{\left\|\boldsymbol{Z}'^{(\ell-1)}\boldsymbol{D}^{1/2}\boldsymbol{Y}\right\|}{\left\|\boldsymbol{Z}'^{(\ell-1)}\boldsymbol{D}^{1/2}\boldsymbol{X}\right\|} \leq (1-\epsilon)\frac{\left\|\boldsymbol{Z}'^{(\ell-1)}\boldsymbol{D}^{1/2}\boldsymbol{Y}\right\|}{\left\|\boldsymbol{Z}'^{(\ell-1)}\boldsymbol{D}^{1/2}\boldsymbol{X}\right\|}. \tag{62}$$

Corollary 11 also confirms that $\left\|\boldsymbol{Z}'^{(L)}\boldsymbol{D}^{1/2}\boldsymbol{X}\right\| > 0$. Using an easy induction, we have

$$\frac{\left\|\boldsymbol{Z}'^{(L)}\boldsymbol{D}^{1/2}\boldsymbol{Y}\right\|}{\left\|\boldsymbol{Z}'^{(L)}\boldsymbol{D}^{1/2}\boldsymbol{X}\right|} \leq (1-\epsilon)^{L-1}\frac{\left\|\boldsymbol{Z}'^{(1)}\boldsymbol{D}^{1/2}\boldsymbol{Y}\right\|}{\left\|\boldsymbol{Z}'^{(1)}\boldsymbol{D}^{1/2}\boldsymbol{X}\right\|}. \tag{63}$$

Therefore,

$$\lim_{L \to \infty} \frac{\left\|\boldsymbol{Z}'^{(L)}\boldsymbol{D}^{1/2}\boldsymbol{Y}\right\|}{\left\|\boldsymbol{Z}'^{(L)}\boldsymbol{D}^{1/2}\boldsymbol{X}\right\|} = 0. \tag{64}$$

Combining above arguments, we obtain that

$$\lim_{L \to \infty} \lim_{n \to \infty} \frac{\left\|\boldsymbol{V}_n^{(L)}\boldsymbol{D}^{1/2}\boldsymbol{Y}\right\|}{\left\|\boldsymbol{V}_n^{(L)}\boldsymbol{D}^{1/2}\boldsymbol{X}\right\|} = \lim_{L \to \infty} \frac{\left\|\boldsymbol{Z}'^{(L)}\boldsymbol{D}^{1/2}\boldsymbol{Y}\right\|}{\left\|\boldsymbol{Z}'^{(L)}\boldsymbol{D}^{1/2}\boldsymbol{X}\right\|} = 0. \tag{65}$$

## D  THE ROLE OF FFN: WHAT MAPPINGS ARE GREAT MAPPINGS

**Lemma 12.** *Let $\sigma : \mathbb{R}^d \to \mathbb{R}^d$ be defined as $\sigma(\boldsymbol{z}) = \boldsymbol{W}\boldsymbol{z}$, where $\boldsymbol{W}$ is a non-singular matrix, then $\sigma$ is a great mapping w.r.t. $(\gamma_{\min}, \gamma_{\max}, \mathbb{R}^d, \boldsymbol{U})$ for any orthonormal matrix $\boldsymbol{U}$, where $\gamma_{\min}$ and $\gamma_{\max}$ are the smallest and largest singular values of matrix $\boldsymbol{W}$, respectively.*

The proof of Lemma 12 is obvious.

**Lemma 13.** *Let $\sigma : \mathbb{R}^d \to \mathbb{R}^d$ be a great mapping w.r.t. $(\gamma_1, \gamma_2, \mathscr{Z}, \boldsymbol{U})$ and $\sigma' : \mathbb{R}^d \to \mathbb{R}^d$ be a great mapping w.r.t. $(\gamma_1', \gamma_2', \mathscr{Z}, \boldsymbol{U})$, then $\sigma_1 \circ \sigma_2$ is a great mapping w.r.t. $(\gamma_1\gamma_1', \gamma_2\gamma_2', \mathscr{Z}, \boldsymbol{U})$.*

The proof of Lemma 13 is obvious.

# E  A GENERALIZED FRAMEWORK THAT IS INDEPENDENT OF DATA GENERATING

In the main paper, Theorem 1 conditions on the specific data-generating process used in Park et al. (2024). This is because we need to match the distribution of the attention connection to each token with their stationary distribution in the data. In this section, we show that, with a slightly stronger assumption on the attention map, a general result of the context-wise convergence can be derived.

**Definition 6.** *If a matrix $\boldsymbol{A} = \{a_{k,j}\}_{k,j \in [n]} \in \mathbb{R}^{n \times n}$ is a balanced attention map with parameter $\psi$, and there is a mapping $f : [c] \to [c]$, such that for all $k, j \in [n]$,*

$$a_{k,j} > 0 \implies x_j = f(x_k), \tag{66}$$

*then we say $\boldsymbol{A}$ reflects $f$.*

Notice that Definition 6 is basically Definition 3 but limits the function $f$ maps each node to only one node, instead of a set of nodes as in Definition 3 (and in this case eq. (4) automatically holds). Although this condition seems stronger, following the same idea used in the main paper, that we can compose multiple attention maps into one, this still represents a large family of allowed attention maps.

Next, we prove a similar result as Theorem 1 under Definition 6 that is independent of the input distribution.

**Theorem 14.** *Suppose $\mathcal{V} = \{\boldsymbol{v}_k\}_{k=1}^n \in (\mathbb{R}^d)^n$ is a stable sequence converging to $\mathcal{Z} = \{\boldsymbol{z}_x\}_{x \in [c]}$ with parameter $\gamma$, and $\boldsymbol{A} = \{a_{k,j}\}_{k,j \in [n]} \in \mathbb{R}^{n \times n}$ is a balanced attention map with parameter $\psi$ that reflects a function $f : [c] \to [c]$. Then $\boldsymbol{A}\mathcal{V}$ is a stable sequence converging to*

$$\mathcal{Z}' = \left\{\boldsymbol{z}_{f(x)}\right\}_{x \in [c]} \tag{67}$$

*with parameter $2\gamma\psi + c(\gamma + 2N)$, where $N = \max_{x \in [c]} \|\boldsymbol{z}_x\|$.*

*Proof.* Let $\boldsymbol{A}\mathcal{V} = \{\boldsymbol{u}_k\}_{k=1}^n$.

If $k \leq c$, we have

$$\left\|\boldsymbol{u}_k - \boldsymbol{z}_{f(x_k)}\right\| \leq \left\|\boldsymbol{u}_k - \boldsymbol{z}_{x_k}\right\| + \left\|\boldsymbol{z}_{f(x_k)}\right\| + \left\|\boldsymbol{z}_{x_k}\right\| \leq \gamma + 2N \leq \frac{c(\gamma + 2N)}{\sqrt{k}}. \tag{68}$$

Below, we only need to consider $k > c$.

We have

$$\left\|\boldsymbol{u}_k - \boldsymbol{z}_{f(x_k)}\right\| = \left\|\sum_{\substack{j \in [k] \\ x_j = f(x_k)}} a_{k,j} \left(\boldsymbol{v}_j - \boldsymbol{z}_{f(x_k)}\right)\right\| \tag{69}$$

$$\leq \sum_{\substack{j \in [k] \\ x_j = f(x_k)}} a_{k,j} \left\|\boldsymbol{v}_j - \boldsymbol{z}_{x_j}\right\| \tag{70}$$

$$\leq \sum_{\substack{j \in [k] \\ x_j = f(x_k)}} \frac{a_{k,j}\gamma}{\sqrt{j}} \tag{71}$$

$$\leq \gamma \sum_{j=1}^{k} \frac{a_{k,j}}{\sqrt{j}}. \tag{72}$$

Now, define $S_{k,j} = \sum_{i=1}^{j} a_{k,j}$ and $S_{k,0} = 0$. Since $\boldsymbol{A}$ is balanced, we have $S_j \leq \frac{\psi j}{k}$. Notice that $a_{k,j} = S_{k,j} - S_{k,j-1}$. Thus we have

$$\frac{1}{\gamma} \left\| \boldsymbol{u}_k - \boldsymbol{z}_{f(x_k)} \right\| \leq \sum_{j=1}^{k} \frac{a_{k,j}}{\sqrt{j}} \tag{73}$$

$$= \sum_{j=1}^{k} \frac{1}{\sqrt{j}} \left( S_{k,j} - S_{k,j-1} \right) \tag{74}$$

$$= \frac{S_{k,k}}{\sqrt{k}} + \sum_{j=0}^{k-1} S_{k,j} \left( \frac{1}{\sqrt{j}} - \frac{1}{\sqrt{j+1}} \right) \tag{75}$$

$$\leq \frac{\psi}{\sqrt{k}} + \frac{\psi}{k} \sum_{j=1}^{k-1} j \left( \frac{1}{\sqrt{j}} - \frac{1}{\sqrt{j+1}} \right) \tag{76}$$

$$= \frac{\psi}{\sqrt{k}} + \frac{\psi}{k} \sum_{j=1}^{k-1} \frac{1}{\sqrt{j}} - \frac{\psi(k-1)}{k\sqrt{k}} \tag{77}$$

$$= \frac{\psi}{k} \sum_{j=1}^{k} \frac{1}{\sqrt{j}} \tag{78}$$

$$\leq \frac{2\psi}{\sqrt{k}}. \tag{79}$$

Thus we conclude that $\left\| \boldsymbol{u}_k - \boldsymbol{z}_{f(x_k)} \right\| \leq \frac{2\gamma\psi}{\sqrt{k}}$ for any $k \in [n]$, which means $\boldsymbol{U}$ converges to $\left\{ \boldsymbol{z}_{f(x_k)} \right\}_{k=1}^{n}$ with parameter $2\gamma\psi$.

$\square$

Using Lemma 7 which is also independent of DGP, we can prove the generalized results for a large family of attention maps by combining multiple attention maps that satisfy Definition 6.

# F  INTEGRATING OTHER TRANSFORMER COMPONENTS

In this section, we discuss how our theoretical results can extend to more complex and realistic Transformer architectures beyond the simplified model described in Algorithm 1. We first note that in our framework, any neuron-wise transformation (i.e., operations that apply independently to the representation of each token) can be absorbed into the definition of the great mapping $\sigma$. This includes FFNs as well as normalizations such as LayerNorm or RMSNorm. Therefore, here we focus here two architectural components not yet discussed: residual connections and multi-head attention.

**Residual Connection**  The most critical step in the proof of Theorem 2 is Lemma 8, which establishes that applying a balanced attention map to a stable sequence results in another stable sequence, and that the attention operates implicitly on the limiting representations to which the sequence converges. Introducing residual connections here is straightforward: with a residual connection, eq. (33) would become

$$\boldsymbol{z}'_x = \left( 1 + \rho_A^{(\ell)} \right) \boldsymbol{z}_x + \rho_B^{(\ell)} \sum_{y \in [c]} \frac{w_{x,y}}{d_x} \boldsymbol{z}_y + \rho_O^{(\ell)} \sum_{y \in [c]} \pi_{\mathcal{G}}(y) \boldsymbol{z}_y + \rho_T^{(\ell)} \boldsymbol{v}_1, \tag{80}$$

which is simply adding 1 to the $\rho_A^{(\ell)}$ coefficient. This modification only affects the $\delta_q$ term in Theorem 2, which becomes

$$\delta_q \left[ (\rho_A + 1)\boldsymbol{I} + \rho_B \boldsymbol{M} \right], \tag{81}$$

which makes the eigenvalue separation ratio smaller. This can slow the convergence, but will not prevent it as long as the eigenvalue separation ratio of $\boldsymbol{M}$ is large enough.

**Multi-head Attention** Lemma 7 shows that any Lipschitz combination of stable sequences remains a stable sequence. Since multi-head attention can be viewed as a Lipschitz combination of multiple single-head attentions, it follows that a multi-head attention mechanism also satisfies Lemma 8, as long as each individual head does. All other parts of the theoretical framework extend accordingly. Notice that the coefficients in Lemma 8 may differ by constant factors in the multi-head case, but this does not affect the asymptotic conclusions in Theorem 2, which only concern limiting behavior.

## G EMPIRICAL VERIFICATION

As discussed in the main text, Theorem 2 relies on a relatively strong structural assumption about the attention maps. It is therefore essential to verify whether these assumptions hold in practice. In this section, we empirically examine this question.

Specifically, in Figure 5, we compute the proportion of A,B,T type attention connections defined in Section 4. Note that Type O connections (i.e., connecting to arbitrary tokens) are excluded from this analysis, as they cover all positions. For each head, we compute the fraction of attention weights (across all layers) that fall into types A, B, or T. Overlapping cases (e.g., the connection from the second token in the sequence to the first one can be considered both as B and as T type) are counted only once. The figure shows that a large proportion of attention weights (> 72% in total) indeed falls into these structured types, lending empirical support to our theoretical assumptions.

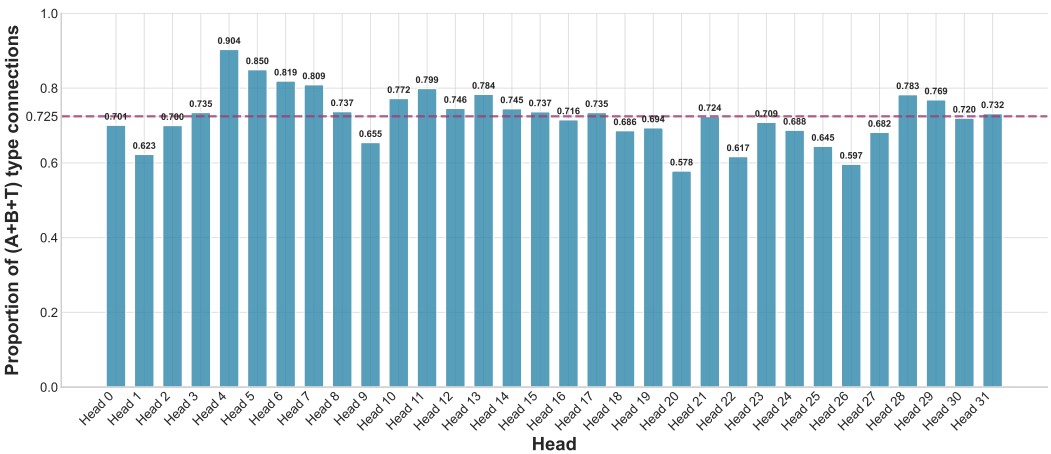

Figure 5: **Proportion of structured attention connections.** For each attention head, we sum attention weights that fall into type A,B and T across all layers, and divide them by total attention weights (which is equal to the number of tokens per layer, since the attention weights are normalized). The dotted horizontal line is the average proportion over all heads.

Moreover, in order to verify the balanced attention assumption (Definition 2), we compute an attention balancedness score for the attention maps. For each attention map $A$, and a position $(k, j)$, where $j < k$, the balancedness score at this position is defined as

$$B_{k,j} = \frac{\sum_{i=2}^{j} a_{k,i}}{\frac{j}{k} - a_{k,1}}. \tag{82}$$

Intuitively, the larger $B_{k,j}$ is, the attention map is more concentrated on the initial part of the sequence, thus imbalanced. Notice that we ruled out $a_{k,1}$ explicitly, as it stands for attention sink, which has been ruled out by the $T$-type attention in our theoretical framework. If we omit the attention sink, then the maximum value of $B_{i,j}$ is essentially the definition of the $\psi$ in Definition 2. The distribution of $\psi$ is presented in Figure 6. It is clear from the figure that the attention balancedness scores are generally small, verifying our balanced attention map assumption.

### G.1 EXTRA VERIFICATION OF THE ROBUSTNESS

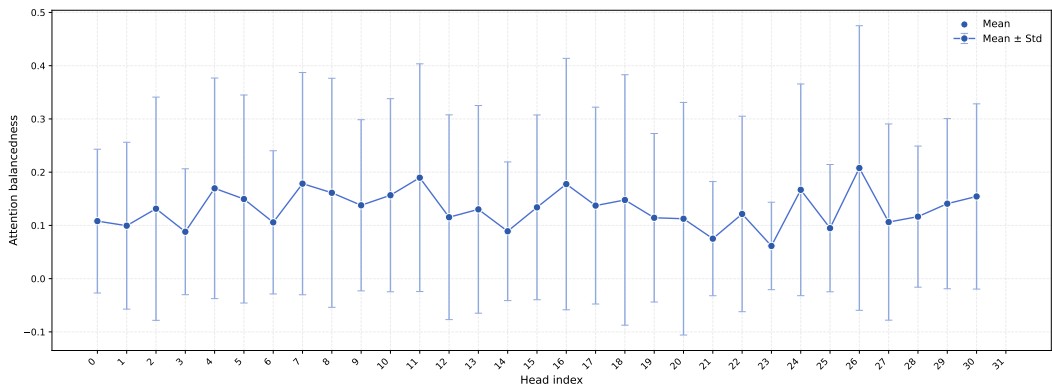

Figure 6: **The attention balancedness score.** For each attention head, we compute all the $B_{i,j}$-s over all positions and layers, and present the mean and standard deviation of their values.

In Section 5.5, we verified our theoretical results by showing that in-context learning is inherently robust to the high-frequency noise. In this sub-section, we present complementary experiment results, which has the same setting as the experiment presented in Section 5.5, but with a higher noise rate in input data. Specifically, we present the results with noise rate 2.5% and 5% in Figure 7 and Figure 8 respectively. The results show that even with a higher input noise, the in-context learning predictions still converge to the noiseless data, which supports our theoretical prediction.

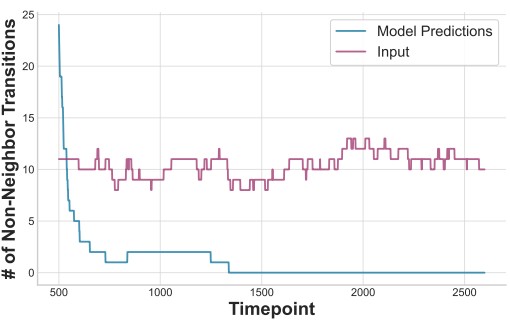 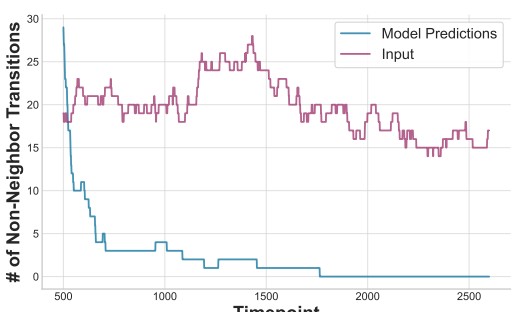

Figure 7: **Predicted Robustness Against Input Noise with 2.5% noise.** We inject 2.5% random noise into the input sequence and plot the number of non-neighbor transitions within a sliding window of the last 500 tokens.

Figure 8: **Predicted Robustness Against Input Noise with 5% noise.** We inject 5% random noise into the input sequence and plot the number of non-neighbor transitions within a sliding window of the last 500 tokens.

## LARGE LANGUAGE MODEL USAGE

In preparing this submission, we used a large language model (ChatGPT) as an assistive tool for language polishing. The model did not contribute to the research ideation, experimental design, data analysis, or the generation of scientific content. All substantive content, results, and conclusions presented in this paper were conceived, written, and verified by the authors, who take full responsibility for the work.

