# OpenReview forum: "Provable Low-Frequency Bias of In-Context Learning of Representations"
_ICLR.cc/2026/Conference — Submitted to ICLR 2026_

### Official Review · Reviewer_WxE9 · 2025-10-31

**Soundness:** 1
**Presentation:** 1
**Contribution:** 1
**Rating:** 2
**Confidence:** 4

**Summary:**

The paper theoretically studies the effect of attention maps on data generated from a graph random walk. Under balancedness and stability assumptions, it is shown that token representations converge over both context and across layers. This shows that tokens become biased towards smooth representations, explaining an in-context learning phenomenon empirically observed in a previous paper.

**Strengths:**

The paper shows that in-context learning of token representations arises a consequence of an intrinsic bias towards low-frequency hidden representations in certain attention models on Markovian data. The theory also suggests an intrinsic robustness towards high-frequency noise, which is confirmed via toy experiments.

**Weaknesses:**

* "Convergence" of representations along the context direction (Theorem 1) is not really interesting. This is simply a consequence of assuming the convergence of weight fractions and representations (Definitions 1,3). Basically, the result is saying that if token $x$ attends to each token type $k\in f(x)$ with relative weight $\pi_k$, then the result of attention is the normalized sum of $\pi_k z_k$ (combined with straightforward error/concentration analysis, e.g., the relative empirical frequency converges to $\pi_k$).

* This analysis also relies on unrealistic assumptions to enable perturbation bounds. Why would the nonzero weights converge to the proportion of tokens (Def.3)? This assumption reduces attention to merely doing a counting operation. While induction heads are one example of such an operation as the authors point out, there is no point to studying its "convergence."

* Layerwise convergence also requires very unnatural assumptions. Although the paper claims their layerwise analysis can cover "FFNs, normalization layers, and other non-linearities" (Section 2.2), the "great mapping" assumption constrains $\sigma$ to be bounded above and below by linear functions, and moreover the only example given (Lemma 12) is a linear function. The only explanation given is "This allows us to insert FFNs or other neuron-wise transformations without disrupting convergence" which feels like this was added merely for the sake of enabling analysis. The spectral gap assumption given in Theorem 2 with no explanation is also very strange, see the last bullet point.

* The assumption that the attention map can be decomposed into A,B,O,T types is justified in Appendix G by checking the proportion of attention weights which fit each type. However, the condition that the weights for each token is a specific type, is much weaker than the condition that the entire attention map is decomposable into a linear combination of the 4 matrices $A^{(A)},A^{(B)},A^{(O)},A^{(T)}$ (which has only 3 degrees of freedom).

* The proof of Theorem 2 is wrong. When applying Corollary 11 in (62), the constant terms should be inversed. Otherwise the condition on line 326 can always be satisfied by choosing $\gamma_1=0$ and becomes vacuous. If this is fixed, the condition becomes that $\lambda_q/\lambda_{q+1}$ is bounded below by some large constant for all $q$, it is unclear why this would hold (moreover, this should not be called the spectral gap, which has a different meaning. Also, it should be made clear that this assumption should hold for all $q$).

**Questions:**

See Weaknesses.

---

> ### Author Response · Authors · 2025-11-24
> **Response to Reviewer WxE9 (Part 1)**
>
> We thank the reviewer for the detailed and careful assessment of our work. We appreciate the reviewer's effort in identifying several important issues, particularly regarding the strength of our assumptions and the clarity of the proof of Theorem 2. We acknowledge that several typos in the submitted version led to an incorrect expression in the proof of Theorem 2, which understandably caused confusion; the correction has been made in the revised manuscript. Below, we provide a consolidated response addressing the reviewer's concerns regarding the assumptions, followed by a clarification and correction of the proof of Theorem 2.
>
> **The assumptions are too strong and unrealistic.**
>
> We understand that one of the reviewer’s main concerns is the perceived strength and realism of several assumptions used in our analysis. We acknowledge that our theoretical development is carried out in an idealized and controlled setting. This choice is deliberate: it enables a fine-grained and fully tractable characterization of the specific ICLR phenomena we aim to explain. As in many prior theoretical studies, such as analyses of model collapse based on linear models with Gaussian data [1] or studies of low-rank bias using deep linear networks [2], simplified setups and specific assumptions are often essential for isolating key mechanisms; Despite their idealized nature, these theories have yielded insights that remain influential within their respective domains.
>
> In the same spirit, our framework, though developed under controlled structural assumptions, captures several qualitative behaviors that are highly relevant for understanding ICL. For instance, it explains the characteristic energy-decay pattern, connects this behavior to the geometry of hidden representations, and identifies a low-frequency bias that directly gives rise to the ICLR phenomenon. All of these explanations are empirically verified and consistent with existing observations. Furthermore, the framework predicts the robustness properties of ICL that we subsequently verify empirically. We believe these results provide meaningful conceptual insight into why such behaviors emerge in practice.
>
> At the same time, the framework is actually not restricted to the most specific form of the assumptions presented in the main paper. In Appendix E, we introduce a relaxed formulation of Definition 3 that does not rely on Eq. (4) and establish Theorem 14 under this weaker setting. Together with Lemma 7, this allows convergence to be analyzed operator-wise and then combined, guaranteeing convergence toward the low-frequency space of some induced operator even when the underlying "graph" is no longer explicit. This shows that the double-convergence framework extends beyond the simplified decomposition used to highlight the ICLR phenomenon in the main presentation.
>
> We also acknowledge that explaining why real attention heads exhibit structures approximating these assumptions remains an open question. Our empirical analysis provides partial evidence: the additional visualizations included in the revised version (Appendix G, Figure 6; newly added content marked in blue) show that real attention patterns exhibit locality consistent with our assumptions. Moreover, similar locality and factorization trends have been documented in numerous empirical studies (e.g., [3, 4, 5]), suggesting that these patterns are not unique to the synthetic ICLR setting.
>
> Overall, while our analysis relies on idealized structural assumptions, these assumptions allow us to obtain detailed and explanatory results for the ICLR phenomenon, and the double-convergence framework provides tools that apply more broadly under relaxed conditions. We hope the clarifications in the revised version help better convey the motivation and scope of our assumptions.

---

> ### Author Response · Authors · 2025-11-24
> **Response to Reviewer WxE9 (Part 2)**
>
> **The proof of Theorem 2 is wrong. When applying Corollary 11 in (62), the constant terms should be inversed.**
>
> We thank the reviewer for pointing this out. The submitted version contains some typos that led to an incorrect expression in the proof of Theorem 2. First of all, in Definition 5, the eigenvalues should be arranged in a non-increasing order (so that $\lambda_q \geq \lambda_{q+1}$). With this corrected ordering, as the reviewer has correctly pointed out, $\lambda_q / \lambda_{q+1}$ should be lower bounded by some constant larger than $1$. We have corrected the typos in the revised manuscript.
>
> We note that the assumption does not need to be held for all $q$, but only needs to be held for the specific $q$ we are interested in. For example, if we are interested in showing that the first two principal components of the hidden representations converge to the 2nd and 3rd eigenvectors (as Section 5 is) of $M$, then we can take $q = 3$. This quantity $\delta_q$ is an intrinsic property of the underlying graph defining the DGP. In the setting of Section 5, this graph is the grid graph, whose third and fourth eigenvalues indeed exhibit a nontrivial separation.
>
> We agree with the reviewer that the term "spectral gap" as used in the submitted version may be misleading. To avoid confusion, we have replaced this terminology with a more precise one. The corrected terminology is now included in the revised version.
>
> ### References
>
> [1] Beyond Model Collapse: Scaling Up with Synthesized Data Requires Reinforcement
>
> [2] Understanding Incremental Learning of Gradient Descent: A Fine-grained Analysis of Matrix Sensing
>
> [3] Modeling Localness for Self-Attention Networks
>
> [4] SparseBERT: Rethinking the Importance Analysis in Self-attention
>
> [5] On the Importance of Local Information in Transformer Based Models

---

### Official Review · Reviewer_Rb1G · 2025-11-01

**Soundness:** 4
**Presentation:** 3
**Contribution:** 2
**Rating:** 4
**Confidence:** 4

**Summary:**

This paper proposes a rigorous theoretical framework to explain the in-context learning of representations (ICLR) phenomenon in Transformers, where the characteristics of the arbitrarily designated token sequence–generating process outweigh the semantic content of the tokens in shaping their representational geometry. By introducing the concept of **Double Convergence**, the authors show analytically that representations first converge over the context to a semantics-independent limiting distribution and then, across layers, exhibit an implicit low-frequency concentration. Using a simplified single-head Transformer with structured attention, they prove that hidden states align with the low-frequency eigenvectors of a graph Laplacian defined by the data-generating process. Through analysis based on this theoretical framework and supporting empirical validations, the paper rigorously formalizes several heuristic observations from the original ICLR study and demonstrates that the ICLR process is robust to noise injection.

**Strengths:**

1. The paper’s theoretical foundation is both conceptually original and mathematically rigorous. The **double convergence**—one with respect to sequence length and another with respect to layer depth—neatly characterizes the representation learning process observed in the original ICLR paper and integrates it (at least up to some simplifications) into the actual forward computation of the model. The authors astutely identify the natural connection between the stationary distribution and transition kernel and the way the attention mechanism operates over embeddings across the context. The decomposition of attention operators into four basic types appears novel and insightful, aligning (to a certain degree) with empirical interpretability findings on the typical modus operandi of attention heads. This formulation also clearly facilitates the spectral analysis of how attention influences model representations.

2. The authors make remarkable efforts to connect their theoretical framework with the anecdotal and heuristic observations in the original ICLR paper. The framework demonstrates strong explanatory power by successfully characterizing where empirical findings align with—or deviate from—the theoretical results, based on well-understood principles from spectral graph theory. It also corrects several ad hoc claims from the ICLR paper through empirical experiments that closely match the a priori theoretical predictions, thereby reinforcing the validity and precision of the proposed theory.

**Weaknesses:**

1. While I acknowledge the authors’ justification in lines 170–178, the settings and assumptions underlying this paper still seem overly restrictive. To obtain context-wise convergence independent of token semantics, the authors assume a single-head attention layer and fixed attention weights—conditions that are far removed from the actual configurations of modern attention modules. Although they empirically validate the factorization of attention weights into four basic forms in Appendix G, which partly supports this assumption, it is highly likely that such simple structures occur only in the highly synthetic ICLR task and do not generalize to more realistic ICL settings involving richer semantics and the complex collective behavior of multiple attention heads (e.g., [1]). Beyond the fixed-weight assumption, the paper further introduces constraints such as balanced attention maps and specific token-mapping behaviors, which make certain theoretical results (particularly Theorem 1) relatively straightforward.

2. The authors restrict their analysis of convergence and low-frequency concentration in ICL hidden states exclusively to the highly constrained ICLR setting. This raises questions about whether their results can generalize to broader ICL scenarios involving representation analysis. For example, the convergence proof requires a context length exceeding ten times the vocabulary size—a reasonable condition for the ICLR task with its small, limited vocabulary but unrealistic for typical ICL tasks using large language model vocabularies on the order of \(10^{4}\) tokens. Moreover, the assumption of semantic-independent convergence diverges from empirical findings in more general ICL contexts, where representational constellations are demonstrably semantic-aware (see [2], [3]).

[1] Singh, Aaditya K., et al. *“What needs to go right for an induction head? A mechanistic study of in-context learning circuits and their formation.”*
[2] Han, Seungwook, et al. *“Emergence and Effectiveness of Task Vectors in In-Context Learning: An Encoder–Decoder Perspective.”*
[3] Kirsanov, Artem, et al. *“The Geometry of Prompting: Unveiling Distinct Mechanisms of Task Adaptation in Language Models.”*

**Questions:**

1, How do you obtain the limiting representations needed to compute the normalized energy scores in Fig 3?

2, In Definition 4, since $Z \in R^{d \times c}, D \in R^{c \times c}, U \in R^{d \times r}$, there seems to be a dimension mismatch in the matrix multiplication

3, In Deifnition 5 you arranged $\lambda_1,...,\lambda_c$ in non-decreasing order, but then in theorem 2 you arranged them instead in a non-increasing order. Consider fixing one of them to ensure consistency.

---

> ### Author Response · Authors · 2025-11-24
> **Response to Reviewer Rb1G (Part 1)**
>
> We thank the reviewer for the thoughtful and detailed assessment of our work, and we appreciate the recognition of the originality, mathematical rigor, and explanatory power of our work. In our understanding, the major concerns from the reviewer concentrate on two points: 1) the assumptions are too strong and seem overly restrictive; 2) the scope is limited to the ICLR setting. In the following, we respond to each of the the concerns.
>
> **The assumption is too strong and seem overly restrictive**
>
> We thank the reviewer for raising this important concern. We agree that some of the assumptions used in our main theoretical development are idealized and may appear restrictive when compared to the flexibility of multi-head attention mechanisms. We would like to emphsize that, the main focuses on the ICLR phenomenon, and some assumptions are introduced to obtain a detailed analysis of the ICLR phenomenon. If we consider broader scenarios, the double-convergence framework can actually be extended beyond the provided assumptions
>
> Specifically, the main critism is that the eq. (4) in Def. 3 might post too strong requirements on the attention map (esstentially requires the overall weight of the attention connecting to a certain token respects the number of times the token shows up in the sequence), and the A,B,O,T decomposition might not be realistic. We here note that these assumptions are essential to make sure the representations converges to the low-frequency space of the underlying graph. In Appendix E, we have provided a relaxed version of Def. 3, which does not require the assumptions in eq. (4), and proved Theorem 14. Theorem 14 and Lemma 7 provide a more flexible formulation in which convergence can be established for each type of attention operator separately and then combined. These results still guarantee convergence to the low-frequency space of some underlying graph, even though, without the structured decomposition in the main paper, the graph on which "frequency" is defined may no longer be explicit. This illustrates that the framework extends beyond the specific synthetic setting used to highlight the ICLR phenomenon.
>
> At the same time, we acknowledge that understanding why real attention heads exhibit structures approximating these assumptions remains an open question, and our empirical analysis only provides partial evidence. To address the reviewer's concerns regarding balanced attention patterns, we have added additional empirical visualizations in the revised version (Appendix G, Figure 6; newly added content marked in blue). These results show that real attention maps exhibit locality patterns that are consistent with the assumptions used in our theory. Moreover, similar locality and factorization patterns have been documented in prior empirical work (e.g. [3, 4, 5]), suggesting that these structures are not unique to the synthetic ICLR setup.
>
>
> Overall, while our theoretical analysis relies on some idealized structural assumptions, we believe the double-convergence framework provides tools that extend more broadly, and that the specific assumptions used in the main paper are essential to capture the detailed phenomena observed in ICLR phenomenon.

---

> ### Author Response · Authors · 2025-11-24
> **Response to Reviewer Rb1G (Part 2)**
>
> **Realisticness of the framework**
>
> We thank the reviewer for raising this important point and for suggesting several relevant references. We fully acknowledge that our analysis is developed in an idealized and controlled setting. This idealized setting is chosen deliberately: it allows us to obtain a fine-grained and fully tractable characterization of several behaviors observed in ICLR. For complex real-world data distributions, many additional factors play a role, and developing a complete theory in such settings remains extremely challenging. This choice is in line with prior theoretical work, which typically adopts idealized settings to isolate key mechanisms. For example, studies of model collapse often use linear models with Gaussian data [1], and analyses of low-rank bias focus on deep linear networks [2]. Despite using simplified settings, these theories are considered useful because they are able to explain many phenomena within their respective domains. Similarly, our theory, despite also focusing on a controlled and idealized setting, still captures several structural aspects that are highly relevant for understanding ICL. For example, our theory explains the observed energy-decay pattern, connects it to the structure of the hidden representations, and identifies a low-frequency bias that directly gives rise to the ICLR phenomenon. Moreover, it predicts robustness properties that we subsequently validate empirically. We believe these insights contribute meaningfully to the broader theoretical understanding of in-context learning.
>
> That being said, several of our assumptions are actually supported by empirical evidence from real LLMs. For example, the balanced attention assumption (Def. 2) essentially requires that the attention weights do not overly concentrate on early tokens (after excluding attention sink, which has been ruled out by the T type). This locality pattern has been reported in many existing work  (e.g. [3, 4, 5]). Regarding the assumption of the semantic-independent convergence (Def. 1), although real representations may not converge exactly to such a form, recent empirical studies (e.g., [6]) show that in-context representations can be decomposed into a predictable component plus a small residual, where the former one is stable and dominates the representation, suggesting that an approximate version of Def. 1 holds in practice.
>
> ### Questions
>
> Below we answer the questions raised by the reviewer.
>
> **1, How do you obtain the limiting representations needed to compute the normalized energy scores in Fig 3?**
>
> They are obtained in the same manner as in Figure 2(c): we collect the hidden representations at the corresponding layer over context positions 2360–2560 and then compute the average representation for each token. We have clarified this procedure in the revised version.
>
> **2, In Definition 4, there seems to be a dimension mismatch in the matrix multiplication**
>
> We thank the reviewer for catching this typo. The shape of ${\boldsymbol{U}}$ should be $c \times p$. We have corrected this typo in the revision
>
> **3, In Deifnition 5 you arranged $\lambda$-s in non-increasing order, but then in theorem 2 you arranged them instead in a non-increasing order. Consider fixing one of them to ensure consistency.**
>
> Thank you for pointing this out. This is a typo: in Theorem 2, the $\lambda$-s are also arranged in non-increasing order. We have corrected this, along with the corresponding instances in the proof.
>
>
>
> ### References
>
> [1] Beyond Model Collapse: Scaling Up with Synthesized Data Requires Reinforcement
>
> [2] Understanding Incremental Learning of Gradient Descent: A Fine-grained Analysis of Matrix Sensing
>
> [3] Modeling Localness for Self-Attention Networks
>
> [4] SparseBERT: Rethinking the Importance Analysis in Self-attention
>
> [5] On the Importance of Local Information in Transformer Based Models
>
> [6] Priors in Time: Missing Inductive Biases for Language Model Interpretability

---

### Official Review · Reviewer_7qor · 2025-11-03

**Soundness:** 2
**Presentation:** 3
**Contribution:** 3
**Rating:** 6
**Confidence:** 2

**Summary:**

This paper provides a theoretical analysis of in-context learning (ICL) focusing on the phenomenon of In-Context Learning of Representations (ICLR) recently observed by Park et al. (2024), where large language models (LLMs) internalize the data-generating process (DGP) of input sequences (e.g., random walks on graphs) within their hidden representations.

The authors introduce a new theoretical framework called double convergence, comprising two intertwined processes:
Context-wise convergence — within each layer, token representations converge to limiting representations determined by token identity.
Layer-wise convergence — across layers, these limiting representations further evolve to capture global structural properties of the DGP.
Together, these processes produce an implicit low-frequency bias in the hidden representations, leading to smooth, globally consistent embeddings that suppress high-frequency (noisy) components.

The paper formalizes these ideas through Theorem 1 (context-wise convergence) and Theorem 2 (layer-wise convergence). This explains several empirical findings in prior work—e.g., why representations align with global graph structure, why energy decays without vanishing, and why models show robustness to high-frequency noise.

Empirical validations (e.g., Figure 2–4) using pre-trained Transformers (Llama-3.1-8B) confirm that model embeddings align with theoretical predictions, exhibiting low-frequency structure and robustness to injected noise.

**Strengths:**

Originality: The paper provides the first rigorous theoretical framework linking ICL dynamics to low-frequency bias and graph-spectral smoothness, a novel perspective that unifies several previously disconnected empirical phenomena (representation alignment, energy decay, noise robustness). The “double convergence” view is both conceptually clean and mathematically tractable.

Quality and clarity:  The work offers formal proofs (Theorems 1–2, Lemmas 6–8) establishing convergence guarantees under balanced attention assumptions, with explicit probabilistic bounds and dependencies on spectral gap and Lipschitz constants. These results extend existing analyses of ICL (e.g., Lu et al. 2024; Li et al. 2025) to deep, nonlinear Transformers—beyond linear attention or single-layer cases. By showing that the limiting representations converge to the low-frequency eigenspace of the Laplacian, the paper provides a principled link between ICL dynamics and graph learning theory. This analogy is insightful and offers a unified way to interpret the “geometry” of learned representations. The alignment between theory and empirical trends strengthens credibility.

Significance: The results deepen theoretical understanding of why LLMs can generalize and remain robust without explicit optimization. The proposed low-frequency bias may become a fundamental explanatory principle for future ICL studies, bridging analysis of Transformers, graph spectral learning, and inductive biases in representation learning.

**Weaknesses:**

1. The analysis assumes that attention maps are “balanced” and externally fixed, depending only on token identity rather than learned representations. While this enables tractability, it significantly limits realism—modern Transformer attention depends on contextual interactions. The authors partially justify this with empirical evidence (“covers 70% of connections”), but the assumption still weakens the generality of the claims.

2. The main theorems are proved under a specific DGP—a random walk on a connected undirected graph with fixed stationary distribution. While extensions are mentioned in Appendix E, the central results remain tailored to this highly idealized setting. The relevance to natural language or general Markov data distributions remains unclear.

3. The empirical validation focuses on graph-based random walks and low-dimensional synthetic setups. No real-world data (e.g., text sequences or few-shot reasoning tasks) are evaluated. This limits the practical relevance of the theoretical insights.

4. The related work discussion is not comprehensive enough. For example, for the mechanism of ICL, it only includes theoretical work  and doesn't mention empirical progress on understanding ICL mechanism.

**Questions:**

How does this framework relate to recent work showing that Transformers perform implicit gradient descent (e.g., Ahn et al., 2023; Von Oswald et al., 2023)? Could low-frequency bias arise as a by-product of gradient-based meta-learning dynamics?

---

> ### Author Response · Authors · 2025-11-24
> **Response to Reviewer 7qor (Part 1)**
>
> We appreciate the reviewer's overall positive evaluation of our work and the thoughtful feedback. Below, we respond to each of the raised concerns in detail.
>
> **The analysis assumes that attention maps are “balanced” and externally fixed, depending only on token identity rather than learned representations. While this enables tractability, it significantly limits realism—modern Transformer attention depends on contextual interactions. The authors partially justify this with empirical evidence (“covers 70% of connections”), but the assumption still weakens the generality of the claims.**
>
> We thank the reviewer for raising this important concern regarding the realism of the balanced attention assumption. While this assumption is introduced to enable a tractable theoretical analysis, we agree that it is important to understand how well it reflects the behavior of real models.
>
> To address this point, we have added an additional visualization of the empirical attention patterns in the revised version (Appendix G, Figure 6; newly added content is marked in blue). The visualization shows that most attention connections do not concentrate on early tokens (except the known attention-sink connections) but instead focus predominantly on local neighbors. This structural pattern is consistent with the balanced-attention assumption.
>
> We hope that these additional results clarify why the assumption captures a meaningful aspect of the empirical attention behavior, while still acknowledging that it is, by design, a simplification used to obtain analytical insights.
>
> **The main theorems are proved under a specific DGP—a random walk on a connected undirected graph with fixed stationary distribution. While extensions are mentioned in Appendix E, the central results remain tailored to this highly idealized setting. The relevance to natural language or general Markov data distributions remains unclear.**
>
> We agree with the reviewer that our theoretical analysis is developed under a specific DGP. This idealized setting is chosen deliberately: it allows us to obtain a fine-grained and fully tractable characterization of several behaviors observed in ICLR. For complex real-world data distributions, many additional factors play a role, and developing a complete theory in such settings remains extremely challenging. This choice is in line with prior theoretical work, which typically adopts idealized settings to isolate key mechanisms. For example, studies of model collapse often use linear models with Gaussian data [1], and analyses of low-rank bias focus on deep linear networks [2]. Despite using simplified settings, these theories are considered useful because they are able to explain many phenomena within their respective domains. Similarly, our theory, despite also focusing on a controlled and idealized setting, still captures several structural aspects that are highly relevant for understanding ICL. For example, our theory explains the observed energy-decay pattern, connects it to the structure of the hidden representations, and identifies a low-frequency bias that directly gives rise to the ICLR phenomenon. Moreover, it predicts robustness properties that we subsequently validate empirically. We believe these insights contribute meaningfully to the broader theoretical understanding of in-context learning.
>
> That being said, several of our assumptions are actually supported by empirical evidence from real LLMs. For example, the balanced attention assumption (Def. 2) essentially requires that the attention weights do not overly concentrate on early tokens (after excluding attention sink, which has been ruled out by the T type). This locality pattern has been reported in many existing work  (e.g. [3, 4, 5]). Regarding the assumption of the semantic-independent convergence (Def. 1), although real representations may not converge exactly to such a form, recent empirical studies (e.g., [6]) show that in-context representations can be decomposed into a predictable component plus a small residual, where the former one is stable and dominates the representation, suggesting that an approximate version of Def. 1 holds in practice.

---

> ### Author Response · Authors · 2025-11-24
> **Response to Reviewer 7qor (Part 2)**
>
> **The empirical validation focuses on graph-based random walks and low-dimensional synthetic setups. No real-world data (e.g., text sequences or few-shot reasoning tasks) are evaluated. This limits the practical relevance of the theoretical insights.**
>
>
> In this work, our goal is to study the ICLR phenomenon in a controlled setting where the underlying mechanisms can be precisely characterized. This allows us to obtain tractable analyses that would be difficult to carry out on noisy natural-language data, where many additional factors interact in complex ways.
>
> At the same time, we note that the ICLR behaviors analyzed in our framework have already been documented in real-world scenarios. For example, [6] empirically validates the ICLR phenomenon on natural-language tasks, and [7] demonstrates the robustness of in-context learning under noisy real data. Our theory is intended to complement these empirical findings by providing a mechanism-level explanation for the patterns they observe.
>
> We view extending our framework to more realistic data distributions as an important direction for future work, and we believe the present analysis provides a useful foundation for such developments.
>
>
> **The related work discussion is not comprehensive enough. For example, for the mechanism of ICL, it only includes theoretical work and doesn't mention empirical progress on understanding ICL mechanism.**
>
> We thank the reviewer for this constructive suggestion. In the revised version, we have expanded Appendix A to include a more comprehensive discussion of empirical studies on the mechanisms of in-context learning.
>
> **How does this framework relate to recent work showing that Transformers perform implicit gradient descent (e.g., Ahn et al., 2023; Von Oswald et al., 2023)? Could low-frequency bias arise as a by-product of gradient-based meta-learning dynamics?**
>
> We thank the reviewer for raising this interesting connection. Yes, our framework is compatible with this implicit optimization perspective. In particular, the layer-wise convergence in our analysis can be interpreted as progressively reducing the "energy" associated with high-frequency components, which resembles an implicit optimization process.
>
> At the same time, our framework differs from these prior approaches in several respects. For instance, our analysis is not restricted to single-layer (as in [8]) or recurrent architectures (as in [9]), and the notion of context-wise convergence enables us to handle models with heterogeneous attention structures across layers. Moreover, our theoretical predictions align with behaviors observed in a real pretrained LLM (Llama-3), without the need to train synthetic models from scratch.
>
> We see our theory as complementary to the implicit optimization interpretations, and we believe that further integrating these viewpoints is an exciting direction for future work.
>
> ### References
>
> [1] Beyond Model Collapse: Scaling Up with Synthesized Data Requires Reinforcement
>
> [2] Understanding Incremental Learning of Gradient Descent: A Fine-grained Analysis of Matrix Sensing
>
> [3] Modeling Localness for Self-Attention Networks
>
> [4] SparseBERT: Rethinking the Importance Analysis in Self-attention
>
> [5] On the Importance of Local Information in Transformer Based Models
>
> [6] Priors in Time: Missing Inductive Biases for Language Model Interpretability
>
> [7] Exploring the robustness of in-context learning with noisy labels
>
> [8] Transformers learn to implement preconditioned gradient descent for in-context learning
>
> [9] Transformers from an Optimization Perspective

---

### Official Review · Reviewer_WNFP · 2025-11-03

**Soundness:** 3
**Presentation:** 4
**Contribution:** 3
**Rating:** 6
**Confidence:** 2

**Summary:**

The paper introduces the concept of Double Convergence as the underlying mechanism behind the  structured representations observed in in-context learning (ICL). Double convergence comprises two parts - context-wise and layer-wise convergence. The paper provides a theoretical framework for these two processes based on some assumptions on the attention maps. It extends the findings of Park et al. (2024) by providing a theoretical framework explaining the empirical observations report by Park et al like the structured hidden representation that capture the structure of the data generating process. It makes reasonable modification to the data generating process from Park et al. by fixing the first set of token in the input sequence. It also verifies the theoretical framework based on simplified transformer model for analysis, where it removed dynamic attention maps and residual connections. Overall, they show that, within the proposed framework, ICL representations converge in two axis resulting in low-frequency bias.

**Strengths:**

1. This studied topic of research is quite relevant for the community. The paper provides a sound theoretical framework to explain the ICLR phenomenon proposing the Double convergence process. Following the defined theoretical framework, it replicates empirical evidence from Park et al. 2024 using the simplified transformer model and further provides new insightful empirical analysis.

2. The detailed energy-decay analysis in hidden representations across layers is novel and support the low-frequency bias hypothesis. Robustness experiment in Section 5.5  adds more evidence for the hypothesis. They also justify the reason for the energy decay using the proposed double convergence framework.

3. The assumptions on simplified transformer model are well supported with empirical verification in Appendix G, showing a good overlap of 72% attention weights.

4. Paper is well-structured and easy to follow. Overall, the theoretical framework is well defined, and empirically validated.

**Weaknesses:**

1. The empirical work shown in this paper is limited to a single DGP process, which is same as Park et al. 2024. Adding experiments on additional DGP process would strengthen evidence for the proposed framework.

2. It would be insightful to see how the model behaves to higher high-frequency noise in input sequence than just 1% in Figure 4/Section 5.5 experiment.

**Questions:**

1. Since this work built on the work by Park et al. 2024. Can you draw parallels with Park et al. 2024 including exact improvements and new insights.

2. Details of the final model architecture used for empirical validation is missing. Can authors  please add more details about the model architecture including number of layers, parameters and also attention pattern visualization (in the next version). Releasing details about the empirical experiments would improve reproducibility.

Minor remarks
- There are a bunch of grammatical errors in the text. Please check the text carefully again.
- Some notations in Algorithm 1 are not defined.

---

> ### Author Response · Authors · 2025-11-24
>
> We thank the reviewer for the overall positive evaluation of our work and the constructive feedback. Below, we provide detailed responses to each of the reviewer's comments.
>
> **The empirical work shown in this paper is limited to a single DGP process, which is same as Park et al. 2024. Adding experiments on additional DGP process would strengthen evidence for the proposed framework.**
>
>
> We would like to clarify that the primary goal of this paper is to develop a theoretical framework for understanding the ICLR phenomenon. The empirical results included in the paper are designed to reproduce and validate the key observations reported in [1], rather than to introduce new empirical findings.
>
> The grid-based DGP used in our experiments is the same setting adopted in [1] and we believe it is sufficiently representative for illustrating the behaviors that our theory aims to explain. Since [1] already provides extensive empirical investigations across multiple DGPs, we chose to focus on this canonical example in order to keep the scope of the present work centered on its theoretical development.
>
> **It would be insightful to see how the model behaves to higher high-frequency noise in input sequence than just 1% in Figure 4/Section 5.5 experiment.**
>
> We thank the reviewer for this helpful suggestion. We have added experiments with a higher rate of high-frequency noise, and the updated results are included in Appendix G and Figure 7 (newly added content is marked in blue in the revised manuscript).
>
> The extended experiments show that as the noise rate increases, the model requires more context length to overcome the stronger high-frequency noise. Nevertheless, its predictions eventually converge to those of the noiseless setting, indicating that the qualitative behavior discussed in Section 5.5 remains robust.
>
> **Since this work built on the work by Park et al. 2024. Can you draw parallels with Park et al. 2024 including exact improvements and new insights.**
>
> We thank the reviewer for raising this important question. Our work indeed builds directly on [1], which provides a careful empirical characterization of the ICLR phenomenon. Our goal is to place these observations on a solid theoretical footing and to uncover the mechanisms underlying in-context learning. Specifically, our contributions extend and complement [1] in the following ways
>
> 1. We introduce the double-convergence framework, which offers a general theoretical lens for analyzing the structure of hidden representations in ICL, which is an aspect not developed in [1].
> 2. We theoretically establish and prove a systematic low-frequency bias in in-context learning, showing that the ICLR phenomenon follows as a direct consequence. In [1], this effect is reported empirically but without a theoretical account.
> 3. We provide a complete theoretical explanation for the observed energy decay: why it decreases yet remains bounded away from zero. [1] documented this behavior empirically, but its underlying cause was not explained.
> 4. Our theory predicts robustness of in-context learning under perturbations, and we validate this prediction with new experiments. This robustness aspect was not examined in [1].
>
> Overall, we believe our work complements [1] by providing theoretical foundations and substantial new insights that explain and generalize their empirical observations.
>
>
> **Details of the final model architecture used for empirical validation are missing. Can authors please add more details about the model architecture, including the number of layers, parameters, and also attention pattern visualization (in the next version). Releasing details about the empirical experiments would improve reproducibility.**
>
> We thank the reviewer for these helpful suggestions on improving the clarity and reproducibility of the empirical setup. As stated in the caption of Figure 2, all experiments are conducted using the Llama-3.1-8B model (provided through the NNSight library). In the revised version, we have made this information more explicit.
>
> Regarding attention-pattern visualization, we think direct heatmaps are difficult to interpret for sequences exceeding 2000 tokens, especially given the presence of noise in real pre-trained models. To address the reviewer's concern, we have added an additional experiment that verifies the balanced attention assumption underlying our theoretical analysis. These additional results, which verify the key structural properties of the attention patterns, are included in Appendix G and Figure 6 (added content is marked in blue in the revision).
>
> We hope these additional details and analyses improve the clarity and reproducibility of our empirical evaluation.
>
> ### References
>
> [1] ICLR: In-Context Learning of Representations

---

### Meta-Review · Area_Chair_7B3b · 2026-01-06

**Summary:**

This paper studies in-context learning of representations under a highly idealized Transformer model and shows that, under strong assumptions, representations first stabilize token-wise and then behave like a low-pass filter on a graph, suppressing high-frequency components. While the phenomenon itself is interesting, the paper does not analyze how such behavior arises from gradient-based training or optimization dynamics. Instead, it observes the behavior and then posits a set of sufficient structural properties under which it would occur. As pointed out most clearly by Reviewer WxE9, the core definitions and assumptions (Definitions 1–5) are very strong and somewhat contrived, and they effectively encode the desired behavior rather than explain its emergence. As a result, the theoretical analysis offers limited insight into why or when real Transformers should satisfy these conditions, weakening the paper’s explanatory contribution.

**Reviewer Concerns:**

The main concerns raised by the reviewers relate to the strength and realism of the assumptions, the lack of connection to training dynamics, and the limited explanatory scope of the theory beyond the idealized setting. These concerns were shared across reviews and articulated most sharply by Reviewer WxE9, whose critique is technically precise and well-founded. While the rebuttal clarifies the intended scope of the work, it does not sufficiently address these foundational limitations.

**Reviewer Scores:**

Given that the central concerns raised by Reviewer WxE9 remain unresolved, it is unlikely that this reviewer would raise their score. Under these circumstances, other reviewers are also unlikely to revise their evaluations upward.

---

### Decision · Program_Chairs · 2026-01-26

Reject